# Dynamic control of endogenous metabolism with combinatorial logic circuits

Felix Moser, Amin Espah Borujeni, Amar N. Ghodasara, Ewen Cameron, Yongjin Park & Christopher A. Voigt*

## Abstract

**Controlling gene expression during a bioprocess enables real-time metabolic control, coordinated cellular responses, and staging order-of-operations. Achieving this with small molecule inducers is impractical at scale and dynamic circuits are difficult to design. Here, we show that the same set of sensors can be integrated by different combinatorial logic circuits to vary when genes are turned on and off during growth. Three *Escherichia coli* sensors that respond to the consumption of feedstock (glucose), dissolved oxygen, and by-product accumulation (acetate) are constructed and optimized. By integrating these sensors, logic circuits implement temporal control over an 18-h period. The circuit outputs are used to regulate endogenous enzymes at the transcriptional and post-translational level using CRISPRi and targeted proteolysis, respectively. As a demonstration, two circuits are designed to control acetate production by matching their dynamics to when endogenous genes are expressed (*pta* or *poxB*) and respond by turning off the corresponding gene. This work demonstrates how simple circuits can be implemented to enable customizable dynamic gene regulation.**

**Keywords** control theory; feedback; metabolic engineering; synthetic biology
**Subject Categories** Methods & Resources; Synthetic Biology & Biotechnology
**Mol Syst Biol. (2018) 14: e8605**

## Introduction

Genetic modifications made to an organism to optimize the production of a chemical or biologic are typically static (Holtz & Keasling, 2010; Brockman & Prather, 2015b). For example, knocking out a gene to redirect metabolic flux implements its impact permanently and continuously (Schellenberger *et al*, 2011). Similarly, introduced pathways are often under unchanging constitutive control (Zhang *et al*, 2002; Burgard *et al*, 2003; Price *et al*, 2004; Schellenberger *et al*, 2011; Morse & Alper, 2016; Deparis *et al*, 2017). While these changes are required to make the product and optimize yield, they

can have a detrimental effect when activated at the wrong time, such as early in growth when resources need to be dedicated to building biomass (San & Stephanopoulos, 1984; Park *et al*, 2007; Michener *et al*, 2012; Brockman & Prather, 2015a; Ceroni *et al*, 2016). Static functions contrast with natural cellular systems that continuously monitor environmental conditions and respond by adjusting gene expression as needed (Shen-Orr *et al*, 2002; Zaslaver *et al*, 2004; Cho *et al*, 2014). Implementing flexible synthetic versions of this regulation would be valuable in engineering projects. For instance, product yields could be optimized by re-balancing enzyme expression to respond to growth phase, the buildup of precursor metabolites, or feedstock concentration (Farmer & Liao, 2000; Liu *et al*, 2015; Zhang *et al*, 2015; Morse & Alper, 2016). Additionally, less external intervention would be required if cells could be pre-programmed to undergo a series of steps during a bioprocess or respond as autonomous agents to bioreactor-borne stresses.

Dynamic gene expression has begun to be implemented in academic metabolic engineering projects (Liu *et al*, 2016; Qian & Cirino, 2016; Min *et al*, 2017; Liu & Zhang, 2018; Zhou *et al*, 2018). These projects depend on genetically encoded sensors that respond to external environmental signals ($O_2$, temperature, pH), the internal cell state (metabolites, growth phase, stress response, redox), the depletion of carbon feedstock (glucose), cell density, or the accumulation of products and by-products (acetate) (Farmer & Liao, 2000; Bayly *et al*, 2002; March & Bentley, 2004; Boccazzi *et al*, 2006; Nevoigt *et al*, 2007; Kang *et al*, 2008; Tsao *et al*, 2010; Liang *et al*, 2011; Michener *et al*, 2012; Zhang *et al*, 2012; Anesiadis *et al*, 2013; Siedler *et al*, 2014; Afroz *et al*, 2015; Liu & Lu, 2015; Soma & Hanai, 2015; Xie *et al*, 2015; Guan *et al*, 2016; Immethun *et al*, 2016; Lo *et al*, 2016; Qian & Cirino, 2016; Rajkumar *et al*, 2016; preprint: Borkowski *et al*, 2017; Bothfeld *et al*, 2017; Gupta *et al*, 2017; He *et al*, 2017; Juarez *et al*, 2017; Klamt *et al*, 2017; Pham *et al*, 2017; Kasey *et al*, 2018). The information transmitted by these sensors can be used to implement feedback control or switch the carbon flux through alternative pathways at the opportune time (Xu *et al*, 2014; Brockman & Prather, 2015b; Liu *et al*, 2015; Ceroni *et al*, 2016). For many products, this approach has been shown to increase yields by maintaining a toxic intermediate below a critical level or separating growth and production phases (Farmer & Liao, 2000; Michener *et al*, 2012; Zhang *et al*, 2012, 2015; Xu *et al*, 2014;

Department of Biological Engineering, Synthetic Biology Center, Massachusetts Institute of Technology, Cambridge, MA, USA
*Corresponding author. Tel: +1 617 617 4851; E-mail: cavoigt@gmail.com

Brockman & Prather, 2015b; Liu *et al*, 2015; Soma & Hanai, 2015; Xie *et al*, 2015; Ceroni *et al*, 2016; Morse & Alper, 2016).

Several strategies can be taken to build such sensors. The ideal sensors consist of a regulator that directly binds to a known signal, such as the metabolite, and then strongly regulates the activity of a promoter (Tang & Cirino, 2011; Zhang *et al*, 2012; Rogers *et al*, 2015; Albanesi & de Mendoza, 2016; Libis *et al*, 2016; Morgan *et al*, 2016; Rogers & Church, 2016). When a sensor for a specific metabolite is unavailable, native promoters that respond to a given stimulus have also been co-opted as sensors (Dahl *et al*, 2013; Yuan & Ching, 2015). However, many native promoters integrate multiple signals, making them respond to alternative or unknown stimuli (Kang *et al*, 2008; Dahl *et al*, 2013; Boyarskiy *et al*, 2016; Rajkumar *et al*, 2016; preprint: Borkowski *et al*, 2017; Hoynes-O'Connor *et al*, 2017; Kasey *et al*, 2018; Siu *et al*, 2018). One approach to address this is to put the operators for a transcription factor into the "clean" background of a constitutive promoter (Cox *et al*, 2007).

A sensor can be genetically modified to change the threshold of signal required to activate it. For example, increasing the expression level of the regulator can make the sensor turn on earlier and mutations can tune the binding constant to the ligand (Nevoigt *et al*, 2007; Moser *et al*, 2013; Afroz *et al*, 2015; Feher *et al*, 2015; Wang *et al*, 2015; Gupta *et al*, 2017; Mannan *et al*, 2017; Landry *et al*, 2018). However, an individual sensor can only implement a switch at a one defined cell state and cannot be used to drive a series of events (Wang *et al*, 2015; Gupta *et al*, 2017). An alternative approach to modifying the sensors is to select a set of sensors that turn on at different times during a bioprocess and then use a genetic circuit that responds to a pattern of sensor activities to turn on at a defined point. During a bioprocess, many conditions change dynamically inside the reactor and inside of individual cells. Therefore, the same set of sensors can be integrated in different ways to generate different dynamic responses.

There is precedent for using genetic circuits to alter a sensor's response (Karig & Weiss, 2005; Slusarczyk *et al*, 2012; Brophy & Voigt, 2014; Hoynes-O'Connor & Moon, 2015). Connecting a sensor to a circuit is simplified when both are transcriptional; that is, when the output of the sensor is a promoter and the inputs/outputs of a circuit are promoters. Circuits have been used to integrate multiple sensors, change their threshold, amplify the response, convert a transient input to a permanent response, and toggle between outputs (Chen & Bailey, 1994; Kobayashi *et al*, 2004; Bennett *et al*, 2008; Moon *et al*, 2011; Wang *et al*, 2011; Moser *et al*, 2012; Solomon *et al*, 2012; Soma *et al*, 2014; Soma & Hanai, 2015; Rantasalo *et al*, 2016; preprint: Borkowski *et al*, 2017; Bothfeld *et al*, 2017; He *et al*, 2017; Ryo *et al*, 2017; Kasey *et al*, 2018).

One way to respond to a pattern of sensor activities is to use genetic circuits that implement logic operations. Combinatorial logic is defined as a relationship in steady state in which the circuit outputs are a function of only the inputs. While circuits themselves do not implement dynamics, when the inputs (sensors) are changing over time, the output of the circuit will also change. Integrating more sensors makes the response more specific to a set of conditions or period of time during growth (Immethun *et al*, 2016; He *et al*, 2017). Larger logic gates can simultaneously integrate many sensors and control multiple output promoters, each turning on in response to a different pattern of sensor activities (Callura *et al*,

2012; Moon *et al*, 2012; Guan *et al*, 2016; Nielsen *et al*, 2016; Green *et al*, 2017).

There are a number of genetic tools to connect the output promoters of a circuit to the control of endogenous or recombinant genes. The output promoter could be used to directly express enzymes (Temme *et al*, 2012; Immethun *et al*, 2016) or orthogonal RNA polymerases that transcribe multi-gene pathways (Temme *et al*, 2012; Segall-Shapiro *et al*, 2014; Bonde *et al*, 2015; Song *et al*, 2017; Harder *et al*, 2018). The output promoter can also be used to turn genes off using CRISPRi or sRNA/RNAi (Drinnenberg *et al*, 2009; Qi *et al*, 2013). These methods have been used to optimize titers by knocking down enzymes of central metabolism at an opportune time or to redirect flux through a heterologous pathway (Callura *et al*, 2012; Solomon *et al*, 2012; Anesiadis *et al*, 2013; Na *et al*, 2013; Oyarzun & Stan, 2013; Soma *et al*, 2014; Brockman & Prather, 2015a; Lv *et al*, 2015; Wu *et al*, 2015; Zalatan *et al*, 2015; Deaner & Alper, 2017; Harder *et al*, 2018; Kasey *et al*, 2018). Proteases have also been developed that target a tag that can be added to an enzyme, though this requires modification of the target enzyme (Cameron & Collins, 2014). The ability to degrade the enzyme pool is critical for rapidly eliminating its activity, particularly when the growth rate is low and proteins are only slowly diluted (Soma *et al*, 2014; Brockman & Prather, 2015a).

In this manuscript, we develop three sensors that respond to generic signals that change over the course of bioproduction and are agnostic to a particular product pathway. Oxygen and glucose sensors are constructed by placing FNR/CRP operators into a constitutive promoter and optimizing for dynamic range using oligonucleotide arrays (Kosuri *et al*, 2010) and fluorescence-assisted cell sorting (FACS). A third sensor that responds to acetate was selected from the literature (Bulter *et al*, 2004) and modified to improve its response. Each of these signals responds at a different time during growth: The low oxygen sensor turns on first, followed by the turning off of the glucose sensor, and finally the acetate sensor turns on. Simulations of many genetic circuits implementing these sensors' signals into different logic operations show that diverse responses are possible. From these, we select several based on layered AND and ANDN gates, construct them, and verify their temporal response. As a proof-of-principle, we design two genetic circuits to respond during periods of endogenous *poxB* and *pta* expression, respectively, as determined using RNA-seq. The circuit controlling *poxB* is turned on during the transition to stationary phase, and the circuit controlling *pta* is turned on early in growth. When the circuits are on, they repress the native genes using a combination of CRISPRi and proteases. The resulting circuits are able to control the appropriate genes at early and late stages of growth, thus reducing acetate accumulation. This demonstrates how different configurations of sensors and gates can be used to generate responses at different times and thereby control carbon flux through endogenous metabolism.

# Results

## Design of glucose, oxygen, and acetate sensors

The simultaneous use of multiple sensors requires that they respond to independent stimuli and do not interfere with each other's response. Further, they require a large dynamic range to facilitate

their connection to circuits. For oxygen and glucose, we and others have built sensors based on native promoters and heterologous transcription factors (Anderson *et al*, 2007; Garcia *et al*, 2009; Immethun *et al*, 2016). However, we were concerned that these would either respond to additional unwanted cellular signals or that their reported dynamic ranges were insufficient. Initially, a number of natural *Escherichia coli* promoters were gleaned from the literature and tested, but their dynamic range proved to be too low (Appendix Fig S1). Therefore, synthetic promoters were designed to respond only to select regulatory proteins and screened variations to identify those that produced a large dynamic range.

The approach to build the glucose and oxygen sensors utilizes a previously published method to generate large libraries of constitutive promoters (Kosuri *et al*, 2013). A library of 11,964 synthetic promoters was computationally designed by varying the promoter backbone and the placement of operators for *E. coli* transcription factors that respond to each signal (Fig 1A). First, twelve constitutive promoter variations were generated, each made up of one of four $\sigma^{70}$-associated promoter sequences (−35 to +1) and one of three randomly generated spacer sequences for the −60 to −35 and +1 to +50 (Fig 1B). Within these sequences, the operators for the glucose- and oxygen-sensing transcription factors were placed at all possible locations (Cox *et al*, 2007; Stanton *et al*, 2014b). For glucose, the operators bind to either the global regulators cAMP receptor protein (CRP; Lawson *et al*, 2004) or FruR (Kochanowski *et al*, 2013), although no promoters with the latter operator ultimately emerged from the screen. For oxygen, the operator is for the fumarate and nitrate reductase (FNR) transcriptional activator, which is directly modified by oxygen via a Fe-S cluster (Constantinidou *et al*, 2006). The full set of promoters was synthesized using a CustomArray oligo array and cloned into a reporter plasmid (p15A origin) upstream of green fluorescent protein (*gfp*). Constitutive expression of a red fluorescent protein (*rfp*) enabled us to correct for variation in copy number of the plasmid (Materials and Methods). RiboJ was included upstream of *gfp* in order to insulate against genetic context effects that occur when it is transcribed from different promoters (Lou *et al*, 2012).

The promoter library was then transformed into *E. coli* MG1655, and FACS sorting was used to screen for activity. For the glucose sensor, cells were grown in the presence of 0.4% glucose and then sorted using a threshold for high GFP:RFP fluorescence (Fig 1A). The recovered variants were then grown in the absence of glucose and re-sorted, this time recovering cells below a threshold GFP:RFP fluorescence. This was repeated for three cycles, after which 95 promoter variants were recovered and tested for their on/off response. The same approach was applied to identify oxygen sensors, where the three FACS cycles were performed by iterating between aerobic and anaerobic growth (Materials and Methods). The top glucose- and oxygen-responsive promoters to emerge from these screens were PgluA7 and PfnrF8, respectively. Their responses were compared to native promoters and the strong constitutive promoter BBa_J23101 (Fig 1C and D; 2016; Kelly *et al*, 2009). The replacement of each sensor's operator with a random sequence eliminated its response (Fig 1C and D). The promoters only respond to their corresponding signal (Fig 1E).

To characterize the promoters as sensors, their response was measured as a function of inducer concentration under conditions that approximate steady state (Materials and Methods). The best

glucose sensor (PgluA7) shows a maximum 18-fold dynamic range and achieves half-maximum induction at 0.1% glucose (Fig 1F). The best oxygen sensor (PfnrF8) produces a 25-fold induction and achieves its half-maximum output at a dissolved oxygen (DO) concentration of 36 μmol/l (Fig 1G). For both promoters, the transition between the off and on states occurs uniformly throughout the population of cells (Appendix Fig S2). The responses of both the glucose and oxygen promoters are rapid, achieving 8-fold and 7-fold activation, respectively, after 1 h (Fig 1F and G, and Appendix Fig S3).

For the acetate sensor, we tested one previously designed by Liao and co-workers based on the PglnAP2 promoter, which responds to phosphorylated NtrC (*glnG*) in a *glnL* knockout stain (Fig 1H; Bulter *et al*, 2004). In our hands, this promoter produces a 16-fold induction in *E. coli* MG1655, but requires knocking out the receptor NtrB (*ΔglnL*; Materials and Methods), which limits its use to strains in which this gene is deleted or repressed. We found that truncating the promoter at the +1 start site (PglnAP2s) improved the dynamic range to 250-fold by reducing the leakiness of the off state (Fig 1H). The half-maximum response occurs at 13.8 mM acetate, and the response to intermediate concentrations is bimodal (Appendix Fig S2). In addition, the response is slower than the other two sensors. It should be noted that the response is sensitive to the pH of the media and changes when other genes are knocked out (Appendix Fig S4; Bulter *et al*, 2004). Because *ΔglnL* knockout mutation interferes with the nitrogen starvation response, we used a nitrogen-rich media and did not observe any growth defects due to this mutation (Appendix Table S1).

The three sensors (PfnrF8, PgluA7, PglnAPs) were tested for orthogonality to each other's signals (low oxygen, glucose, acetate; Fig 1E). The three sensors are activated by their cognate stimuli, with minimal measurable cross-reactivity between the acetate and glucose sensors (Appendix Fig S5). Thus, they can all be used together within one circuit, although some care needs to be taken to avoid crosstalk.

The three sensors were then evaluated in shake flask experiments where cells were seeded into a defined glucose-based media common in industry (Moser *et al*, 2012) and grown for over 24 h (Materials and Methods). For these experiments, GFP was fused to a degradation tag to better measure off-times (McGinness *et al*, 2006). Glucose, dissolved oxygen (DO), and acetate were monitored throughout growth by offline liquid chromatography and an oxygen sensor probe (Materials and Methods). Glucose and DO decrease over time due to cell growth and metabolism (Fig 2A and B, Appendix Fig S6). The inoculum culture is first grown without glucose, but when cells are added to glucose-containing media (*t* = 0 h), the glucose sensor rapidly turns on and remains on until glucose is consumed after 15 h (Fig 2A). The DO sensor turns on to the absence of oxygen, which is consumed during growth, causing the sensor to turn on after 8 h (Fig 2B). Acetate accumulates late in growth and the sensor turns on when the acetate concentration passes the 15 mM threshold after 14 h (Fig 2C and Appendix Fig S7).

## Sensor integration with combinatorial logic circuits

Over the course of a growth experiment, the output of the three sensor promoters is continuously changing. These promoters can be connected as inputs to a logic circuit that responds only when each

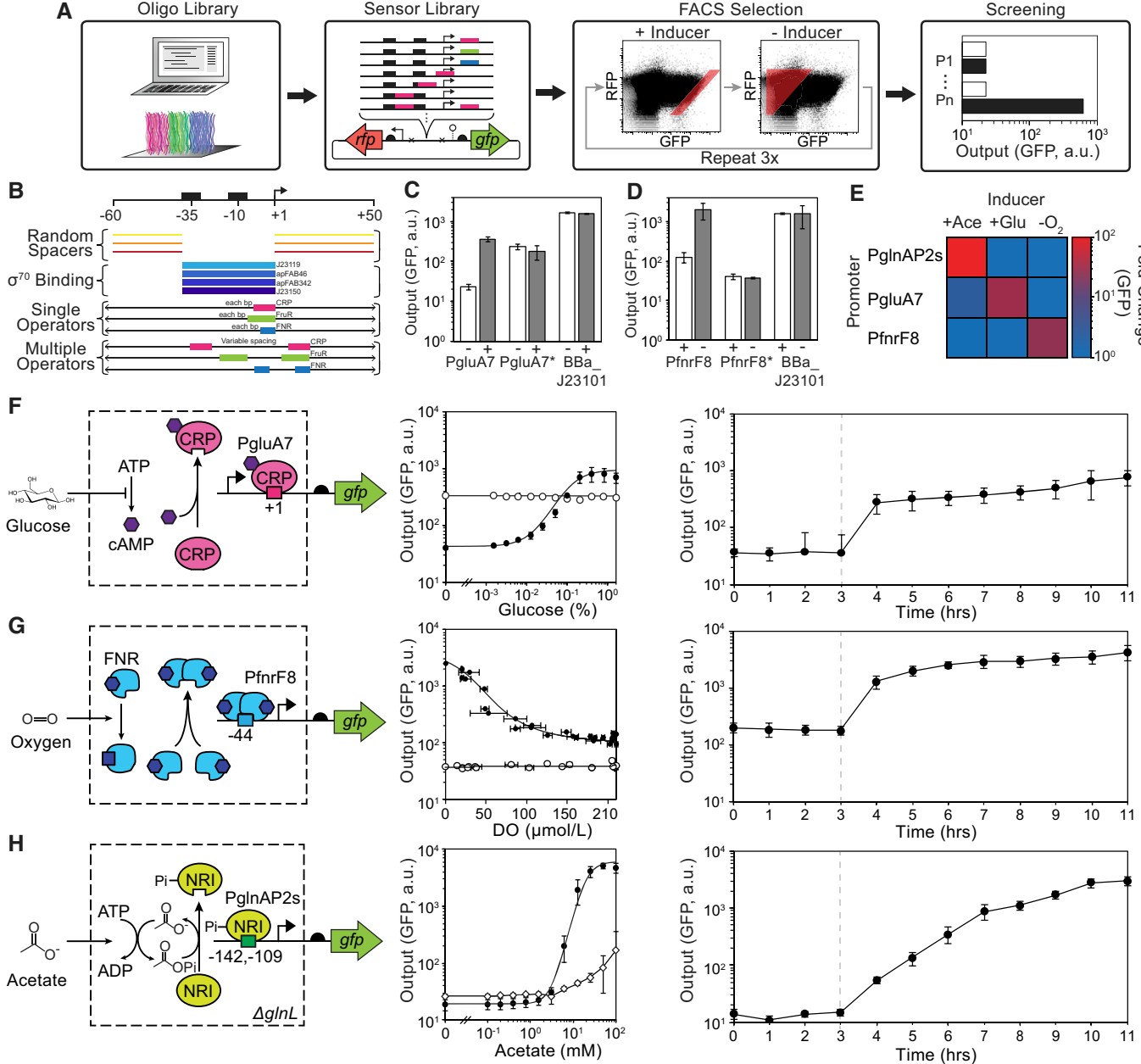

**Figure 1.  Design and optimization of glucose, acetate, and oxygen sensors.**

A    Scheme for sensor design including (left to right): computational design and DNA chip oligo library synthesis, insertion into a plasmid reporter, FACS enrichment, and screening of select mutants in the presence (black) and absence (white) of stimulus. All three sensors were synthesized on a single chip (indicated by the colors).

B    Design of the promoter library. The location, number, and name of promoter elements are shown. The permutations included different constitutive core promoters (blue) flanked by random spacers (orange). Single operators (colored bars) were varied across the ranges shown with single nucleotide resolution. When two operators were included, they were inserted at multiple sites and the distance between them was varied by up to 6 bp from each site.

C    Shown are the responses of the glucose sensor promoter (PgluA7), a negative control lacking the CRP operator (PgluA7*), and a constitutive promoter (BBa_J23101) to the presence (+) and absence (−) of glucose.

D    Shown are the responses of the oxygen sensor promoter (PfnrF8), a negative control lacking the FNR operator (PfnrF8*), and a constitutive promoter (BBa_J23101) to the presence (+) and absence (−) of oxygen.

E    The orthogonality of the sensors is shown. The averages and standard deviations for these data are provided in Appendix Fig S5.

F–H  Shown are the schematics and responses for the glucose, oxygen, and acetate sensors, respectively. The response functions (center) are shown for each sensor (black circles) compared to promoter variants where the operators are removed (PgluA7*, PfnrF8*; open circles). Horizontal error bars in the PfnrF8 response reflect one standard deviation of three dissolved oxygen (DO) measurements. For the acetate sensor, the response of the sensor is shown in unmodified *Escherichia coli* MG1655 with *glnL* intact (open diamonds). The dynamics of induction are shown (right graph) where cells are induced at the time indicated by the dashed line (see text). Representative cytometry florescence distributions for Fig 1F–H are shown in Appendix Fig S2.

Data information: Error bars represent one standard deviation of three independent experiments done on different days.

sensor is at the correct level. Thus, by connecting the sensors to circuits that implement different logic operations (truth tables), the circuits will produce different responses over time. Because the circuits are based on the layered expression of regulators (a cascade), different circuits that encode the same truth table can result in different dynamics due to delays in signal propagation. To determine the range of possible dynamics, simulations were run for all possible 3-input logic circuits designed based on layered AND, ANDN, and NOR gates (Moon *et al*, 2012; Nielsen *et al*, 2016; Appendix Fig S8). The inputs to the circuit are the empirically measured output promoter activities of the three sensors over time (Fig 2A–C). The circuit response is simulated using a simple set of ordinary differential equations (ODEs) to model the on- and off-times of each gate (Materials and Methods). The simulation results for the full set of circuits are shown in Appendix Fig S8, of which a subset of characteristic responses are shown in Fig 2D. Circuits are predicted to yield a diverse range of dynamic behaviors with varying on- and off-times.

Two modeled circuits were built and tested experimentally (Fig 2E and H). The circuits are built using AND gates that utilize an activator (InvF) that requires the expression of a second protein (SicA) to turn on an output promoter (PsicA; Appendix Fig S9). In addition, the repressor PhlF is used to build ANDN gates (Stanton *et al*, 2014a). The first circuit (Fig 2E–G) has three inputs and two outputs, where each output is designed to respond to a different combination of signals, and thus, each will turn on and off at different times. In this circuit, the glucose and acetate sensors drive the SicA/InvF system to compose an AND gate. The oxygen sensor drives the repressor PhlF to turn off a second copy of the glucose-inducible promoter to compose AND NOT (ANDN) logic that activates in the presence of glucose AND NOT low oxygen. The second circuit (Fig 2H–J) is based on a three-input, one-output logic gate that implements (A and B) AND NOT (C) logic, where the C signal (low $O_2$) turns off the gate by expressing PhlF, which represses the expression of both InvF and SicA. These gate designs were selected to reduce delays that can occur due to layering and the likelihood of a transiently incorrect response (fault) occurring (Hooshangi *et al*, 2005; Mangan *et al*, 2006; Moon *et al*, 2012).

The circuits were constructed and their responses measured over time. For the first circuit, outputs 1 and 2 were measured using a degradation-tagged GFP and RFP, respectively (Fig 2E and F). Their responses to changes in the sensor activities are in accordance with the encoded logic (Fig 2G). As predicted, Output 2 turns on early in growth and Output 1 turns on late in growth (after an initial transient response around $t = 0$ due to the shift from glycerol to glucose). This demonstrates that a single circuit can encode multiple responses by integrating the same set of input sensors; for example, to turn one process off and another on at defined times during growth. The second circuit shows strong induction under the correct conditions and produces a temporal pulse in the activity of the output promoter (Fig 2H–J). The fluorescence distributions from cytometry show that the circuit responses are monotonic (Appendix Fig S10).

### Dual transcription/translation control over output genes

The output promoter can be used to drive the expression of a gene. It is more complicated when the goal is to turn a gene off. Tools such as CRISPRi can be used to repress the transcription of a gene by expressing a sgRNA that targets dCas9 to block its transcription (Qi *et al*, 2013). However, it does not rid the cell of mRNA/protein that has already been expressed, which will continue to be active until they slowly degrade. This will be problematic when a rapid response is required (e.g., to eliminate an enzyme to redirect metabolic flux). Methods to induce the degradation of mRNA and proteins include the transcription of sRNA and targeted proteolysis (Na *et al*, 2013; Ghodasara & Voigt, 2017). Here, we evaluated different combinations of CRISPRi/sRNA/proteases in order to evaluate their impact on the magnitude and timing of the knockdown of a gene (Fig 3A).

First, the three methods were evaluated for their ability to reduce the expression of RFP encoded on a plasmid (Materials and Methods). For CRISPRi, dCas9 was expressed from a weak constitutive promoter and sgRNA was expressed from a DAPG-inducible PphlF promoter encoded on plasmid (p15a origin). The sgRNA encodes a previously published 20 bp targeting sequence that binds to the non-template strand early in the mRFP1 CDS (Qi *et al*, 2013). This system represses RFP expression by 69-fold (Fig 3B). The sRNA sequence is designed to bind a "barcode" sequence (Ghodasara & Voigt, 2017) adjacent to the *rfp* ribosome binding site (RBS). The sRNA transcript is expressed from an IPTG-inducible promoter, and

**Figure 2. Response of sensors and circuits during growth in batch experiments.**

All of the temporal responses shown on the left-hand side of this figure were measured under identical experimental conditions (Materials and Methods).

A–C   The responses of the glucose, oxygen, and acetate sensors during growth in shake flasks are shown. The colored lines (right axes) correspond to the measured changes in the stimuli (Materials and Methods). The colored bars under (C) show the times when the output promoter of each sensor should be on, based on the response functions shown in Fig 1.

D   Simulations of circuit dynamics. Examples of different characteristic responses are shown, selected from the full set of simulated circuits (Appendix Fig S8). The lines shown in bold blue colors correspond to those circuits experimentally tested. The simulated output promoter activities are in relative promoter units (RPU; Nielsen *et al*, 2016).

E   The responses of a 3-input, 2-output circuit are shown.

F   Shown are the circuit (left) and genetic diagram (right) of the circuit corresponding to (E). In the genetic diagram, the dashed line and * indicates a second copy of the PgluA7 promoter that drives *rfp* expression and is repressed by PhlF via an immediately downstream PhlF operator.

G   The response of the circuit in (E, F) to different combinations of stimuli under the same conditions as Fig 1 (Materials and Methods). The (+) and (−) indicate whether the output promoter of each sensor is active under those conditions. Bars where the circuit is predicted to be on are shown in gray and white when predicted to be off.

H   The response of a 3-input 1-output circuit is shown.

I   Shown are the circuit (left) and genetic diagram (right) of the circuit corresponding to (H).

J   The response of the circuit in (H, I) to different combinations of stimuli under the same conditions as Fig 1 (Materials and Methods).

Data information: Representative cytometry florescence distributions for (A–C and E–J) are shown in Appendix Figs S7 and S10, respectively. Error bars represent one standard deviation of three independent experiments done on different days.

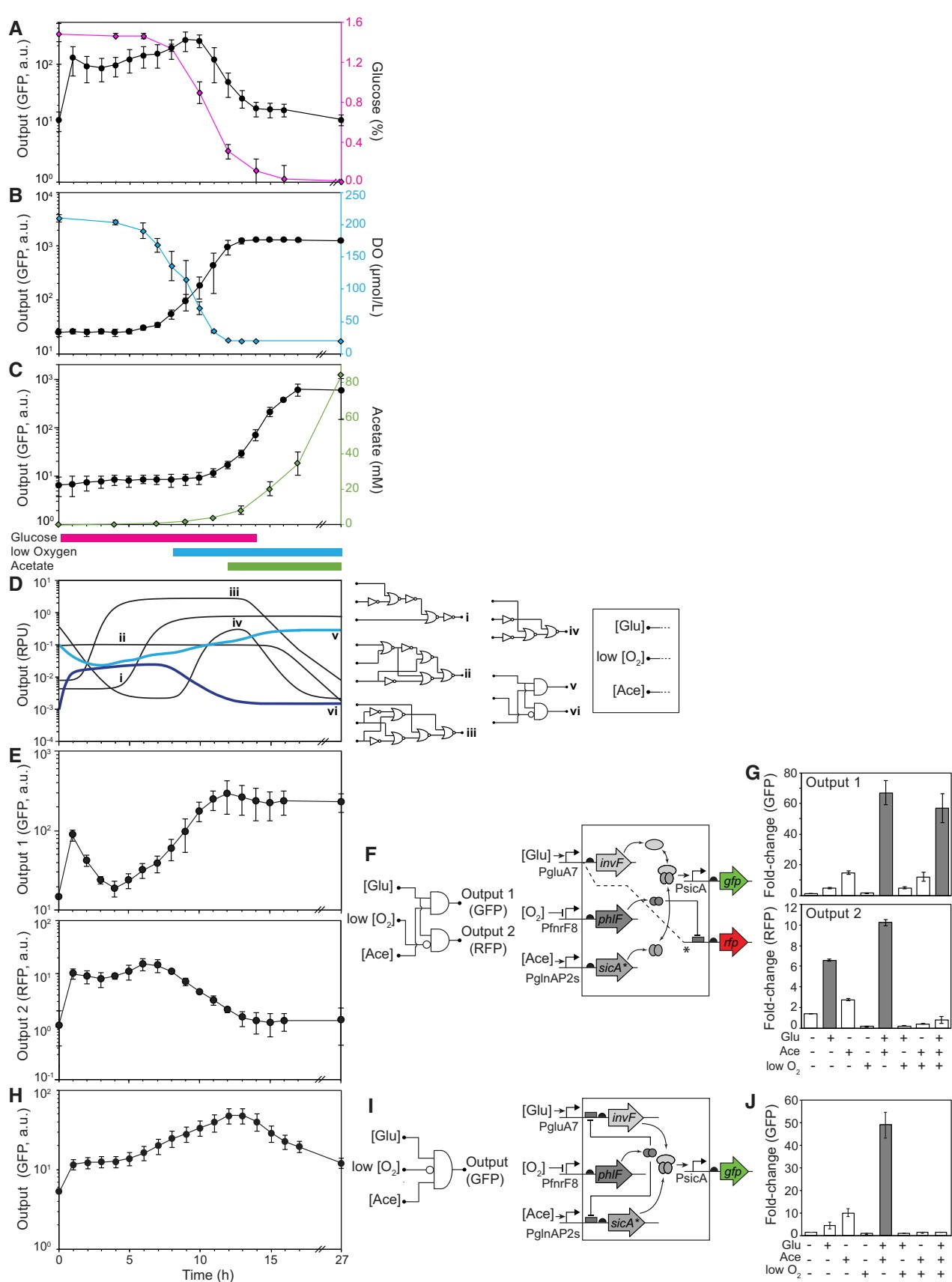

**Figure 2.**

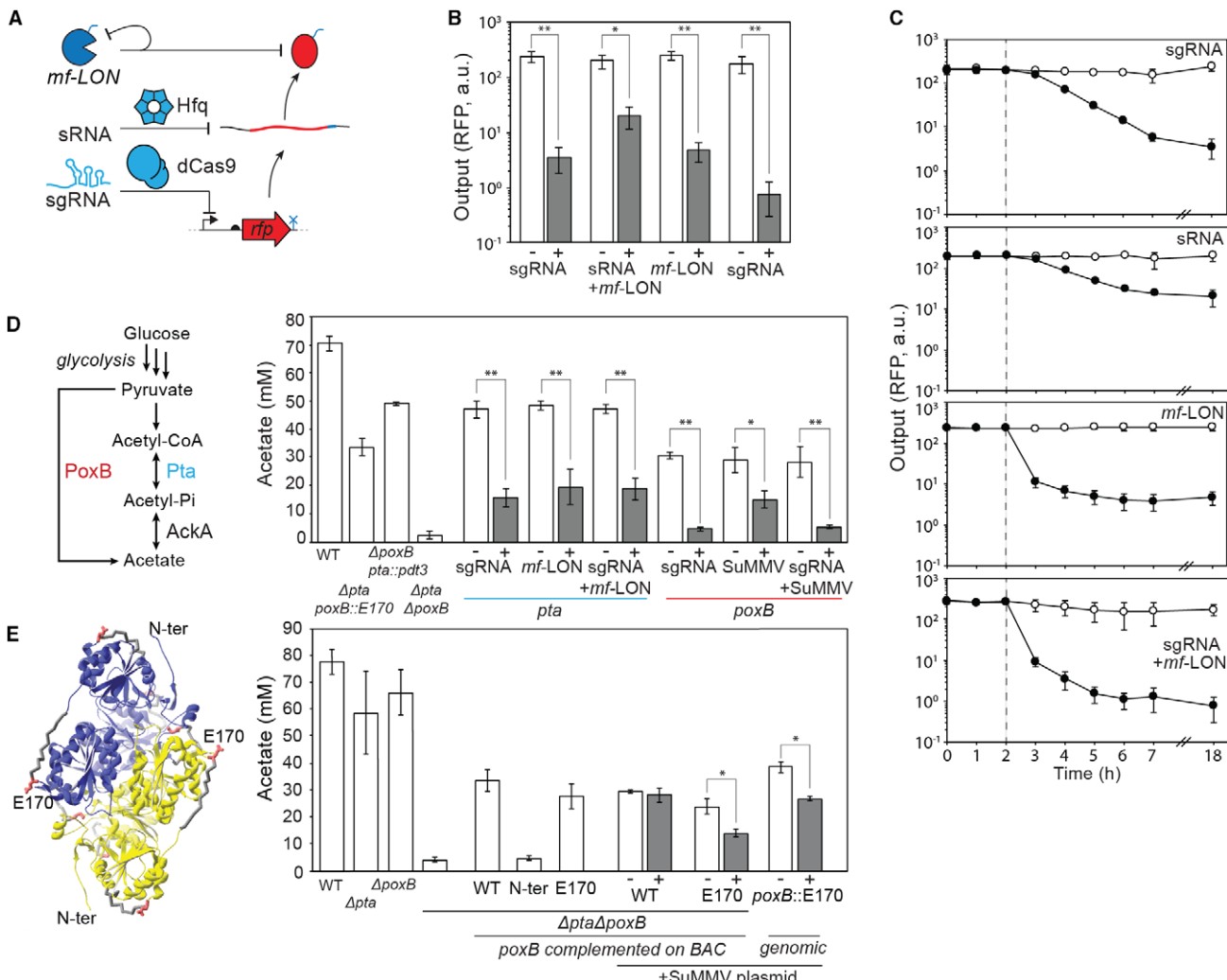

**Figure 3. Eliminating protein activity using a combination of CRISPRi, sRNA, and proteases.**

A   Schematic showing three levels of repression. A small guide RNA (sgRNA) directs deactivated Cas9 (dCas9) to block transcription from a promoter. sRNA binds to the mRNA and promotes degradation by recruiting Hfq. The *mf*-Lon protease targets a tag (blue) added to the protein. The protease is also targeted to itself to reduce toxicity (Materials and Methods).

B   Reduction of fluorescence of RFP by the different mechanisms of repression after 18 h of growth (Materials and Methods). The inducers are either 1 mM IPTG (sRNA) or 25 μM DAPG (sgRNA, *mf*-LON). sgRNA and *mf*-LON are co-transcribed on a single transcript that is processed by ribozymes (Appendix Fig S13).

C   The dynamics of repression by each of the mechanisms is shown. Empty circles are uninduced and black circles are induced (1 mM IPTG or 25 μM DAPG) at the 2-h time point (dashed line; Materials and Methods).

D   Metabolic pathways to acetate in *Escherichia coli* are shown along with the targeted enzymes. The impact of either knocking out these enzymes or knocking them down by the various mechanisms after 20 h is shown. On the left of the graph are shown empty strains containing either no acetate pathway modifications (WT; MG1655Δ*glnL*), a double deletion (Δ*pta*Δ*poxB*), or single knockouts and protease tag modifications (Δ*poxB pta::pdt3*, Δ*pta poxB::E170*). On the right are shown knockdowns of *pta* (blue) and *poxB* (red), tested in strains Δ*poxB pta::pdt3* and Δ*pta poxB::E170*, respectively. Knockdowns are generated by expression of sgRNA, proteases, or both mechanisms encoded on a single transcript.

E   Design of a PoxB mutant that can be targeted by the SuMMV protease. The structure of PoxB is shown (PDB: 3ey9; Neumann *et al*, 2008), highlighting the amino acid sites selected for degron insertion (in red). The impact on acetate production of different acetate pathway mutants is shown on the left of the graph. On the right of the graph, a double knockout mutant (Δ*pta*Δ*poxB*) contains variants of *poxB* expressed from a BAC. Note that complementation of unmodified *poxB* (WT) from the BAC does not fully reconstitute acetate production. N-ter and E170 refer to the location of the SuMMV degron tag in *poxB*. A plasmid expressing protease SuMMV from a DAPG-inducible promoter was also introduced into strains containing *poxB* variants WT and E170 complemented on a BAC and a strain *poxB* with an E170 site replacement on the genome (poxB::E170). The graph shows the acetate produced in these strains when either no inducer (white) or 25 μM DAPG (grey) is added to the culture.

Data information: Representative cytometry fluorescence distributions for (B, C) are shown in Appendix Fig S11. Error bars represent one standard deviation of three biological replicates done on different days. Bars with single (*) and double (**) asterisks indicate a statistically significant difference with *P*-values < 0.01 and 0.001, respectively, as assessed by an unpaired *t*-test.

a ribozyme is inserted immediately after the promoter to cleave the 5′-UTR to ensure it is active (Ghodasara & Voigt, 2017). This sRNA system represses RFP expression by 10-fold in the presence of IPTG (Fig 3B). Because sRNA is less effective in our hands than CRISPRi and showed no improvement in dynamics (Fig 3C and Appendix Fig S11), we selected CRISPRi to combine with proteolysis.

The *Mesoplasma florum* LON (*mf*-LON) protease was tested for its ability to degrade a target protein containing the corresponding 27-amino acid C-terminal tag (Materials and Methods; Cameron & Collins, 2014). This tag was fused to RFP and *mf*-LON is expressed from an IPTG-inducible promoter. Some toxicity was observed in the initial designs, which could be mollified by using a weak RBS and fusing the protease to its own degradation tag to enable auto-proteolysis (Appendix Fig S12). The expression of *mf*-LON is less effective than CRISPRi (Fig 3B), but its impact occurs rapidly with a significant reduction in the first hour (Fig 3C).

We then tested if co-expression of CRISPRi and protease could behave synergistically to enable more rapid and potent knockdown of RFP. To co-express both from the same output promoter, a transcript was designed that contains the sgRNA and the protease, separated by a ribozyme (RiboJ; Fig 3A, and Appendix Figs S11 and S13). Although we could not detect greater potency of the knockdown compared to sgRNA alone, we observed a rapid knockdown that matched the speed of the *mf*-LON knockdown (Fig 3B and C).

### Targeting genome-encoded enzymes for repression

Core metabolic enzymes involved in acetate production were then targeted using the dual CRISPRi/*mf*-LON system. Acetate overproduction late in growth reduces product yield and can be toxic at concentrations above 1 g/l (Wolfe, 2005; Eiteman & Altman, 2006; De Mey *et al*, 2007). However, it is beneficial to the cell when glucose is plentiful because it facilitates the recycling of coenzyme A and balances the redox state; thus, knocking out the producing genes can be detrimental under these conditions (De Mey *et al*, 2007). The primary route to acetate production is *pta* and *ackA*, which are expressed constitutively and have been reported to be upregulated in low oxygen conditions (Shalel-Levanon *et al*, 2005). Pyruvate dehydrogenase (*poxB*) can also make acetate directly through pyruvate decarboxylation and FAD reduction and is expressed in stationary phase (Chang *et al*, 1994). Therefore, we selected *pta* and *poxB* as the genes to target with dual CRISPRi/protease control.

When *E. coli* MG1655*ΔglnL* is grown for 20 h in 1.6% glucose, the resulting culture contains ~ 70 mM acetate (Fig 3D; Materials and Methods). Knocking out either *pta* or *poxB* reduces the rate of acetate accumulation during growth but has little effect on the final acetate concentration (Appendix Fig S15). The dual knockout of *pta* and *poxB* results in 10-fold less acetate (Fig 3D and Appendix Fig S15). Deletion of *ackA* is severely detrimental to growth (Appendix Table S1).

We tested the impact of knocking down *pta* and *poxB* with CRISPRi on acetate production. For this, *pta* and *poxB* knockdowns were tested in *E. coli* MG1655*ΔglnLΔpoxB* and *E. coli* MG1655*ΔglnLΔpta*, respectively (Fig 3D; Materials and Methods). Three sgRNAs were designed to target the non-template strand near the beginning of each gene and were found to have approximately the same effect. One of these sgRNA sequences was selected for each gene, and the corresponding acetate reductions are shown in Fig 3D.

The genes for *pta* and *poxB* were then tagged such that different proteases could be used to target their degradation. The N-terminal of Pta is critical for function, but its C-terminus can be functionally tagged (Campos-Bermudez *et al*, 2010). Therefore, *mf*-LON pdt3 tag was fused to the C-terminus of *pta* in its native context in the genome, generating the strain *E. coli* MG1655*ΔglnLΔpoxB pta*::pdt3. The method by which the tag is introduced into the genome leaves only a short FLP recombinase scar after the tag, as previously described (Cameron & Collins, 2014). *Escherichia coli* MG1655*ΔglnL pta*::pdt3 showed no change in growth rate or acetate productivity compared to the wild-type strain (Appendix Table S1). However, the expression of *mf*-LON in this strain reduces acetate production (Fig 3D).

Both the N- and C-terminal ends of PoxB are critical for function, making it ineligible for *mf*-LON degradation (Neumann *et al*, 2008; Weidner *et al*, 2008). Therefore, the *Potyvirus* SuMMV protease was selected, which cleaves peptide bonds immediately after the sequence EEIHLQ (Fig 3E; Fernandez-Rodriguez & Voigt, 2016). Cleavage of this sequence can be used to expose the N-terminal degron sequence FLFVQ (Bachmair *et al*, 1986; Tasaki *et al*, 2012; Fernandez-Rodriguez & Voigt, 2016). A screen was developed to identify an internal site that could accommodate replacement of the native sequence with EEIHLQFLFVQ without disrupting function. PoxB variants were made that replaced the sequence at positions E170–180, E325–335, E347–357, and E469–479 (Fig 3E, Appendix Figure S14). These four internal *poxB* variants as well as an N-terminal fusion variant were expressed from their native promoter from a bacterial artificial chromosome (BAC) in *E. coli* MG1655*ΔglnLΔptaΔpoxB* (Materials and Methods). As expected, the N-terminal fusion of the tag eliminated PoxB activity (Fig 3E). A maltose binding protein (MBP)-SuMMV fusion was expressed from a DAPG-inducible promoter on a separate plasmid (Materials and Methods). One variant (E170–180) showed a reduction in acetate that is both protease- and tag-dependent (Appendix Fig S14). We then modified the genomic copy of *poxB* in *E. coli* MG1655*ΔglnLΔpta* with the integrated SuMMV cleavage site and N-degron, generating the strain *E. coli* MG1655*ΔglnLΔpta poxB*::*E170* (Materials and Methods). When SuMMV was expressed in this strain, there is a similar decline in acetate production (Fig 3E).

### Design of circuits that target *pta* and *poxB* when they are transcribed

Two circuits were designed that integrate environmental signals and subsequently knock down genomically encoded *poxB* and *pta* to reduce acetate (Fig 4). Previously, it has been reported that *pta* contributes the most to acetate production during exponential growth and that *poxB* contributed only after entry into stationary phase (Chang *et al*, 1994; Wolfe, 2005). To confirm this, we performed transcriptomic studies with *E. coli* MG1655*ΔglnL*. We grew this strain in shake flasks in minimal media containing 1.6% glucose for 27 h and assessed the *pta* and *poxB* transcript abundance using RNA-seq at six time points (Materials and Methods). These data confirmed that *poxB* expression peaks during the transition from exponential to stationary phase, whereas *pta* expression is higher during exponential growth (Fig 4B and E; Appendix Fig S15).

A circuit was selected to regulate *poxB* by integrating the glucose and acetate sensors using an AND gate (Fig 4A). The predicted

response matches when *poxB* is transcribed (Fig 4B). Note that this circuit implements closed loop feedback control as acetate production is downregulated when sensed. The output of the circuit is used to drive the expression of the sgRNA targeting *poxB* as well as the SuMMV protease that degrades the tagged enzyme. The background transcription from PsicA was initially too high when the circuit was off, which necessitated its mutation to reduce activity (Materials and Methods). When parent cells lacking the circuit (*E. coli*

MG1655Δ*glnL*Δ*pta poxB::E170*) are grown, 70 mM acetate accumulated after a day of growth in media containing 1.6% glucose (Fig 4C; Materials and Methods). When the circuit is included, the acetate accumulates normally until the time in which the circuit is expected to become active and then slows its accumulation, subsequently reducing the final acetate concentration by half. A control was constructed in which the sgRNA is targeted to RFP and was tested in a strain in which no protease tag was fused to *poxB* (*E. coli*

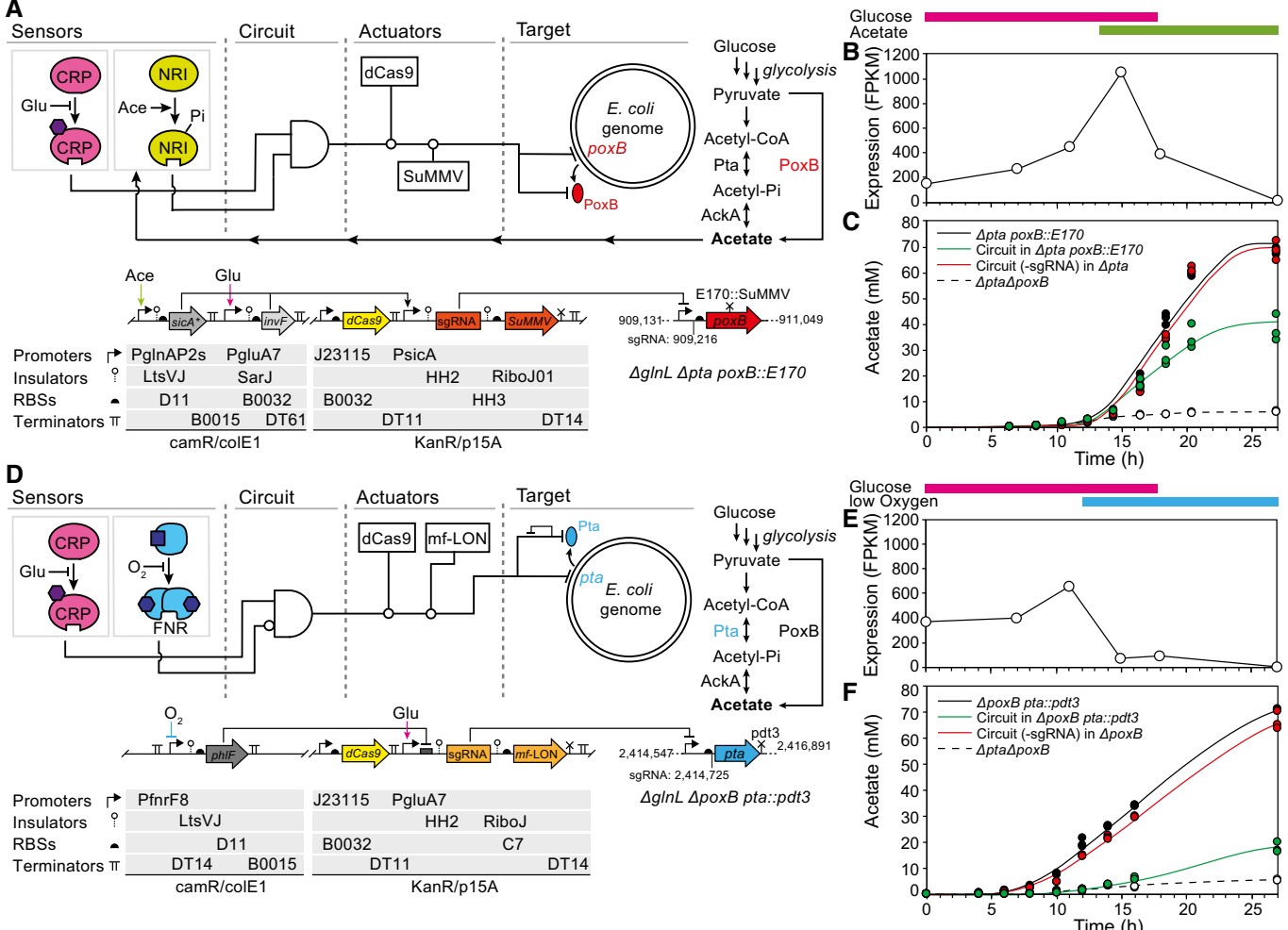

**Figure 4.  Dynamic control of acetate production.**

A    The genetic circuit designed to control *poxB* is shown. The strain genotype is shown, including the positions targeted by the sgRNA and the SuMMV protease.

B    The transcription of *poxB* over time (*Escherichia coli* MG1655Δ*glnL*), as calculated from RNA-seq data (Materials and Methods). The colored bars indicate the times at which the glucose/acetate sensors will be on based on metabolite measurements. It should be noted that these are right-shifted compared to Fig 2 due to slower growth of the tested strains.

C    The production of acetate is shown over time for *E. coli* MG1655Δ*glnL*Δ*pta poxB::E170* (black) as compared to the same strain containing the complete circuit (green). A version of the circuit in which the sgRNA is targeted to *rfp* (not present in the system) and is tested in MG1655Δ*glnL*Δ*pta* (containing an untagged *poxB*) is shown as a control (red). Empty circles connected by the dashed line represent MG1655Δ*glnL*Δ*pta*Δ*poxB*.

D    The genetic circuit design to control *pta* is shown.

E    The transcription of *pta* over time (*E. coli* MG1655Δ*glnL*), as calculated from RNA-seq data (Materials and Methods). The colored bars indicate the times at which the glucose/oxygen sensors will be on based on metabolite measurements.

F    The production of acetate is shown over time for *E. coli* MG1655Δ*glnL*Δ*poxB pta::pdt3* as compared to a strain containing the complete circuit (green) tested in same strain. A version of the circuit in which the sgRNA is targeted to *rfp* (not present in the system) and is tested in MG1655Δ*glnL*Δ*poxB* (containing an untagged *pta*) is shown as a control (red). Empty circles connected by the dashed line represent MG1655Δ*glnL*Δ*pta*Δ*poxB*.

Data information: In (C, F), three experiments were performed and all individual data points are plotted. The lines are fit to averages of these data. Growth curves for each strain are shown in Appendix Fig S16. The corresponding plasmid maps and parts sequences for (A, D) are provided in Appendix Fig S17 and Table S2.

MG1655Δ*glnL*Δ*pta*). This control yielded the same amount of acetate as the empty control strain (Fig 4C).

To control *pta*, a different circuit was chosen that responds in the presence of glucose AND NOT low oxygen (Fig 4D). This is predicted to be on during exponential phase and to turn off as cells transition to stationary phase, which mimics when *pta* is transcribed (Fig 4E). The circuit was constructed by using PfnrF8 to drive the expression of PhlF, which then blocks transcription from the PgluA7 promoter. This promoter serves as the output of the circuit and is connected to the transcription of sgRNA targeting *pta* and the protease *mf*-LON. The output promoter initially showed too much basal transcription, which had to be reduced by mutation (Materials and Methods). When the parent strain lacking the circuit (*E. coli* MG1655Δ*glnL*Δ*poxB pta*::pdt3) is grown in 1.6% glucose for 27 h, it produces 70 mM acetate under the same growth conditions as described above. The same strain containing the circuit greatly reduces acetate accumulation during exponential growth, yielding a 4-fold reduction in acetate after 27 h (Fig 4F). As a control, we tested a circuit in which sgRNA targets RFP and tested it in a strain in which no protease tag is fused to *pta* (*E. coli* MG1655Δ*glnL*Δ*poxB*). This control produced the same amount of acetate as the empty control strain, indicating that the reduction in acetate we observed in the *pta*-targeting circuit is due to circuit outputs.

## Discussion

Digital logic, defined where the signals exist in either a low (0) or high (1) state, is a powerful abstraction because it enables computer algorithms to design the circuit. When applied to genetic circuits, this sometimes invokes criticism that controlling gene expression requires intermediate levels and dynamic control (Zaslaver *et al*, 2004; Young *et al*, 2018), neither of which are embedded within a combinatorial logic circuit itself. However, these circuits can implement dynamic responses by continuously monitoring and responding to changes in the sensor inputs. Further, while the logic minimization algorithms generate a wiring diagram based on Boolean logic, the implementation of the circuit with repressors is in practice fuzzy logic, where the gate responses are intermediate levels that can be tuned through the selection of parts.

In this manuscript, we have demonstrated how combinatorial logic circuits can be used to respond to signals that change over the course of growth and can be used to execute a temporal response. Further, it can be used to implement feedback control, where the signal that forms the closed loop (in this case, acetate) is external to the circuit. Extending the logic diagram to include multiple outputs enables the same sensors to be used to implement varied control over different genes. Here, we chose three generic signals that change over the course of production and a greater degree of control could be achieved by expanding this to include additional sensors (metals, ammonia, pH, redox, toxins), light control, and promoters that respond to stress (Brekasis & Paget, 2003; Dixon & Kahn, 2004; Weber *et al*, 2005; Krulwich *et al*, 2011; Moser *et al*, 2013; Wang *et al*, 2013; Fernandez-Rodriguez *et al*, 2017). As the circuits get larger, one can imagine implementing complex control over a metabolic pathway by turning on different portions of the pathway at different times, coordinating stress responses, and staging a process to

including steps before (seeding and growth) and after (biomass recycling or disposal) the production phase. Examples of advanced approaches include the following: (i) only expressing oxygen-sensitive enzymes under anaerobic conditions (Burgard *et al*, 2012; Immethun *et al*, 2016), (ii) transiently responding to localized stresses within larger bioreactors (Bylund *et al*, 1998; Enfors *et al*, 2001), (iii) lysis system for product recovery (Borrero-de Acuna *et al*, 2017), (iv) flocculation for sedimentation for biomass removal and inhibition of cell growth (You *et al*, 2004; Izard *et al*, 2015), and (iv) elimination of synthetic DNA before disposal (Caliando & Voigt, 2015; Chan *et al*, 2016). Such approaches have been implemented individually, but one can envision linking many together into one large system. Thus, the advantages of the resulting synthetic regulatory control encompass far more than the simplistic, albeit important, concepts of improving titer and yield.

There are some key challenges that must be addressed before large synthetic regulatory networks can be practically implemented. Foremost is the problem of toxicity and stability. Even medium-sized synthetic circuits (≥4 regulators) can slow growth (Sleight *et al*, 2010; Chen *et al*, 2013; Xu *et al*, 2014; Ceroni *et al*, 2018). This can cause instability in the form of plasmid loss or mutations to the genome (Fernandez-Rodriguez *et al*, 2015). Further, the slowing of growth can be devastating for bioproduction. Even when genes only have a slight impact individually, these impacts are additive and quickly compound. This limited the size of the complete systems that we could build to the relatively small circuits shown in Fig 4. Larger circuits that integrated all three sensors, multiple AND/NOT gates, and CRISPRi/protease control of both PoxB/Pta proved to be unstable and significantly slow growth (not shown). Addressing this challenge will require understanding the mechanisms underlying regulator toxicity, quantitative metrics for the resource utilization of regulatory circuitry (e.g., ATP consumption), encoding the circuits in the genome, and redesigning gates to minimize their impact.

## Materials and Methods

### Strains

All cloning and plasmid propagation was carried out in *E. coli* NEB10 (F– *mcr*A Δ(*mrr-hsd*RMS-*mcr*BC) Φ80*lacZ*DM15 DlacX74 recA1 *end*A1 *ara*D139 Δ(*ara leu*) *7697gal*U *gal*K*rps*L*nup*G λ–; NEB #C3019). Promoter measurement plasmids were built on the backbone of pSB3K3, containing a p15A origin of replication and kanamycin resistance marker (2016; Shetty *et al*, 2008). Genomic modifications were generated in *E. coli* MG1655 using the technique of Datsenko and Wanner (Datsenko & Wanner, 2000). For gene knockouts, only the translated CDS (ATG..TAA) of each gene was removed. Fusion of the *mf*-LON pdt3 tag to genome-encoded *pta* was performed as previously described (Cameron & Collins, 2014). Briefly, a PCR product containing the protease tag, a kanamycin resistance marker, and 50 bp of sequence homologous to the target site were integrated into the genome by lambda red recombination. Following identification of a successful modification by PCR, this cassette was transduced by bacteriophage P1 into *E. coli* MG1655Δ*glnL*Δ*poxB*. The kanamycin resistance marker was then removed by FLP recombinase expressed from pCP20 (Datsenko &

Wanner, 2000), which left a 46 bp FLP recombinase scar sequence downstream of *pta*. For the *poxB* modification, we modified pKD4 (Datsenko & Wanner, 2000) with the SuMMV-targeted *poxB* sequence (E170) inserted downstream of a chloramphenicol resistance marker flanked by two FLP recombinase sites (pFM1165; Appendix Figure S17). Using this plasmid, we generated an amplicon by PCR that contained the chloramphenicol marker, part of the *poxB* sequence containing the modification, and 50 bp of sequence homologous to the target site upstream of the native *poxB*. This amplicon was transformed into an *E. coli* MG1655 strain expressing lambda red recombinase (Datsenko & Wanner, 2000). Following screening by PCR of a successful insertion, this cassette was transduced into *E. coli* MG1655Δ*glnL*Δ*pta* with bacteriophage P1. The chloramphenicol resistance marker was then removed by FLP recombinase expressed from pCP20 (Datsenko & Wanner, 2000), leaving a 46 bp scar site immediately upstream of the native *poxB* promoter. All modified genomic loci were verified by PCR and subsequent Sanger sequencing (Quintara).

## Media

Cultures were grown in LB-Miller broth (BD #2020-05-31) for cloning and plasmid propagation. For all other experiments, cultures were grown on a defined industrial minimal medium (MM) containing the following: 5 g/l $(NH_4)_2SO_4$ (Millipore #AX1385-1), 5 g/l $K_2HPO_4$ (VWR #0705), 30 g/l 2-(*N*-morpholino) ethanesulfonic acid (MES; Sigma #M2933), a carbon source as indicated, and a proprietary mixture of micronutrients as previously described (Moser *et al*, 2012). Carbon sources included glycerol (VWR #97062-832) or glucose (BDH #8005-500g). Concentrations of glycerol and glucose here are given as % mass (g/g). Following preparation, the pH of the minimal media was adjusted to 6.8 with NaOH and HCl and the media was filtered through a 0.2 μm filter (Corning #430049). The inducers anhydrotetracycline (aTc; Sigma #37919), 2,4-diacetylphloroglucinol (DAPG; Santa Cruz Biotechnology #12161-86-6), sodium acetate (Sigma #127-09-3), and isopropyl β-D-1-thiogalactopyranoside (IPTG; Sigma #367-93-1) were added to the concentrations indicated. Antibiotics were added at the following concentrations to maintain plasmids in all liquid cultures and plates unless otherwise indicated: 50 μg/ml kanamycin (GoldBio #25389-94-0), 50 μg/ml spectinomycin (GoldBio #22189-32-8), 50 μg/ml ampicillin (GoldBio #69-52-3), and 35 μg/ml chloramphenicol (GoldBio #25-75-7).

## Sensor library design and construction

We automated the design of a library of 11,964 unique 150 bp promoter sequences using a program written in MATLAB (Mathworks). The program concatenates sequences of four different promoters (BBa_J23150, BBa_J23119, apFAB46, apFAB342; 2016; Mutalik *et al*, 2013) with consensus operators of FruR (GCTGA AACGTTTCAAG; Saier & Ramseier, 1996), CRP (AAATGTGATC TAGATCACATTT; Lawson *et al*, 2004), and FNR (TTGATTTACA TCAA; Constantinidou *et al*, 2006), flanking randomized spacer sequences (Appendix Table S2), and 20 bp sequences at each end for amplification (Lawson *et al*, 2004; Kochanowski *et al*, 2013). Individual operator sequences were inserted at every position of each promoter, and multiple operator sequences were inserted at

either the +1 site, the 17 bp spacer between the −10 and −35 sites, or immediately upstream of the −35 site. The completed library sequences were screened for > 5 bp repeats, −10 and −35 near-consensus sequences, and BsaI sites. The library was synthesized as single-stranded oligonucleotides on a microarray chip, cleaved from the chip, and delivered as a desiccated sample (CustomArray). Library oligonucleotides were rehydrated and amplified by PCR for 20 cycles with Phusion DNA polymerase (NEB, #M0530) using the primers oFM1004 (GATTACAGGTCTCTCAGGAAACTCAACTCCT GTGGCGTG) and oFM1005 (GATTACAGGTCTCTCTGCTTTCGCAC GTATACGTGAGTGG) to generate BsaI cut sites with unique 4 bp overhangs on the ends of the product strands. The plasmid pFM436 was amplified with the primers oFM974 (GATTACAGGTCTCA GCAGAAGCTGTCACCGGATGTGCTTTCCGGTCTGATGAGTCC) and oFM976 (GATTACAGGTCTCTCCTGATGCCACCTGACGTCTAAGAA ACCATTATTATCATGACATTAAC) to generate a linear fragment with two BsaI cut sites at each with unique 4 bp overhangs. The PCR products was then gel purified (Zymo #D4001), and library inserts were cloned into pFM436 using the MoClo assembly protocol (Werner *et al*, 2012) such that library promoters were directionally inserted immediately upstream of sfGFP. The resulting promoter library in pFM436 was then transformed into electrocompetent *E. coli* MG1655 for screening. To assess the library quality, we sequenced the inserts of 89 different colonies by Sanger sequencing and found that 35 (39%) contained inserts exactly matching members of the library.

## Sensor FACS screening

To enrich for promoters with the largest dynamic range in response to inducing conditions, we performed repeated cycles of positive and negative screening using fluorescence-assisted cell sorting (FACS). For this, we scraped ~ 150,000 individual colonies from library transformation plates and grew the pooled cells in MM containing 0.4% glycerol for 18 h. Cells were then diluted 1/1,000 into fresh MM containing either 0.4% glucose or 0.4% glycerol for the glucose sensors. For oxygen sensors, cells were grown in MM containing 0.4% glycerol in either aerated tubes or tubes from which oxygen had been removed (see below). Cells were grown for 3 h in these conditions before diluting them to an $OD_{600}$ of 0.01 in phosphate-buffered saline (PBS; VWR # EM-6505). FACS screening was performed on a FACSAria 2 (Becton Dickinson) at the Koch Institute Swanson Biotechnology Center Cytometry Core facility (Cambridge, MA). Cells were sorted based on gates drawn diagonally in the GFP versus RFP plot to correct for variation in plasmid copy number (Elowitz *et al*, 2002). Briefly, a positive screening was performed by collecting at least 1 million cells showing the greatest 5% of GFP to RFP ratio following growth on MM containing 0.4% glucose or no oxygen. Negative screening was performed for all sensors by collecting at least $10^6$ cells showing the lowest 20% of GFP to RFP ratio following growth on MM containing 0.4% glycerol and grown under aerobic conditions. In both positive and negative screening rounds, cells were sorted into 1 ml of LB broth to recover for 3 h and were subsequently plated on LB agar and diluted back 1/100 for overnight growth on MM containing 0.4% glycerol. Each FACS screen was performed identically, excepting alternating growth on positive and negative screening conditions.

## Sensor plate screening

After each round of positive and negative FACS screening, some of the cells were diluted and plated on LB agar. Of the resulting colonies, 95 were randomly selected and screened for GFP induction in response to glucose or anaerobic conditions during growth in 96-well deep-well plates. As a control, an identical construct containing constitutive promoter BBa_J23101 was also tested to measure the effect of induction conditions on constitutive σ70 promoters. For these assays, colonies were picked into 500 μl MM containing 0.4% glycerol in a 2-ml deep-well 96-well plate and grown for 18 h at 37°C in a Microtron plate incubator (Infors) at 1,000 RPM. The cultures were then diluted 1/100 in fresh 500 μl MM containing 0.4% glycerol in two separate 2-ml deep-well 96-well plates. Following inoculation, each plate was covered with a transparent BreatheEasy membrane (USA Scientific #9123-6100) and cultures were grown at 37°C in a Microtron plate incubator (Infors) at 1,000 RPM for 6 h prior to induction. For the glucose sensor screen, one plate was induced by adding 0.8% glucose to each well. For the oxygen sensor, one plate was induced by placing it in a vinyl anaerobic chamber (Coy Type C) containing 2% hydrogen, 98% nitrogen, and a palladium catalyst and shaking it at 37°C in a shaker incubator (VWR #12620-930) at 800 RPM to prevent settling. All plates were grown for an additional 6 h prior to sampling the cultures for cytometry to enable production of the GFP output (see below). Sensor performance was evaluated based on the ratio between the median GFP fluorescence of the uninduced cells compared to the induced cells. The promoters that showed the strongest response were sequenced, sub-cloned into pFM438 (Appendix Fig S17), and further characterized in *E. coli* MG1655Δ*glnL*.

## RBS and promoter library design and screening

RBS library sequences were computationally generated using the RBS Library Calculator Version 1.2 (Salis Lab; Tian & Salis, 2015). Each library contained at least 50 RBS sequences that evenly spanned 2–3 orders of magnitude in calculated strength (arbitrary units). For promoter libraries, base pairs were randomized at the −10 and −35 sites as indicated. These sequences were then integrated into primers for amplification of the target plasmid. Following generation of the DNA library, electrocompetent cells were transformed with the library and plated on LB agar containing antibiotics. Individual colonies were then picked and screened by fluorescence 96-well plate assays.

## Fluorescence assays

The fluorescence responses of sensors and circuits were tested in 96-well deep-well plates (USA Scientific #1896-2000) or 14-ml culture tubes (Falcon #352059) unless otherwise indicated. Fresh cultures were inoculated from single colonies streaked on LB agar from a glycerol stock frozen at −80°C. Inoculum cultures were grown for 6 h in 3 ml of LB media at 37°C in 14-ml culture tubes and then diluted into 3 ml of MM containing 0.8% glycerol in 14-ml culture tubes for 18 h of overnight growth. The initial $OD_{600}$ of this overnight culture was calculated such that after 18 h of growth, the $OD_{600}$ of the culture would not exceed 0.5 to prevent anaerobic induction of the oxygen sensor by a dense culture state

(Appendix Fig S6). The glucose and acetate sensor induction curves were performed in 96-well deep-well plates. For characterization in plates, overnight cultures were diluted to $OD_{600}$ = 0.01 in 500 μl MM containing 0.8% glycerol in a 2-ml deep-well 96-well plate. Following inoculation, the plate was covered with a transparent BreatheEasy membrane (USA Scientific #9123-6100) and cultures were grown at 37°C in a Microtron plate incubator (Infors) at 1,000 RPM for 6 h prior to induction. Oxygen sensors and oxygen-modulated circuits were tested in 14-ml culture tubes, unless otherwise noted, in order to maintain anaerobic conditions following initial removal of oxygen by vacuum and nitrogen cycling as described below. RFP knockdown time courses (Fig 3) were also performed in tubes. For characterization in tubes, overnight cultures were diluted to $OD_{600}$ = 0.01 in 3 ml of MM containing 0.8% glycerol in 14-ml culture tubes. Cultures were then grown in an Innova 44 shaking incubator (New Brunswick) at 250 RPM and 37°C for 6 h prior to induction. Acetate and glucose sensors were induced with glucose and acetate dissolved in MM as indicated. Oxygen sensor cultures were induced by sealing the culture tubes with rubber stoppers (Fisher Scientific #FB57879) and using a vacuum manifold to remove the air from the tube and replace it with nitrogen to ambient pressure. Vacuum and nitrogen cycling was done three times to maximize the removal of oxygen from the headspace. Following induction, cells were sampled at the indicated time points for cytometry analysis, using syringes to penetrate the rubber stoppers for the anaerobic oxygen sensor cultures. Sampled cells were diluted to an $OD_{600}$ of 0.01 or below in cold PBS containing 0.5 mg/ml of Kanamycin and were left at 4°C for at least 1 h prior to cytometry.

## Acetate knockdown assays

CRISPRi, sRNA, and protease knockdown of targeted *pta* and *poxB* were tested in 14-ml culture tubes. Knockdowns of *pta* and *poxB* was tested in *E. coli* MG1655Δ*glnL*Δ*poxB pta*::pdt3 and *E. coli* MG1655Δ*glnL*Δ*pta poxB*::170, respectively. Inoculum cultures were grown for 6 h in 3 ml of LB media at 37°C in 14-ml culture tubes. Cultures were then diluted 1/1,000 into 3 ml of MM containing 0.8% glycerol in 14-ml culture tubes for 18 h of overnight growth. Following overnight growth, inoculum cultures were diluted to an OD600 of 0.01 in 4 ml of MM containing 1.6% glucose and either inducers or no inducers. CRISPRi systems were induced with 25 μM DAPG and 4 ng/ml aTc, sRNA systems were induced with 1 mM IPTG, and proteases *mf*-LON and SuMMV were induced with 25 μM DAPG. Cultures were then grown in an Innova 44 shaking incubator (New Brunswick) at 250 RPM and 37°C. Following overnight growth, 1 ml of each culture was added to a separate 1.5 ml tube and centrifuged at 21,000 *g* for 5 min. This was repeated twice more, keeping half the volume of supernatant after each spin. After the third centrifugation, 100 μl of supernatant was pipetted into a U-bottom 96-well plate (Corning #3797). This plate was covered with an AluminaSeal (Diversified Biotech #ALUM-1000) and kept at 4°C until analysis by liquid chromatography.

## Shake flask cultures

We assessed sensor and circuit performance in shake flask cultures, a common intermediate during industrial scale-up. For these

experiments, we grew cultures of 30 ml MM in 250-ml unbaffled shake flasks (Pyrex No. 4980) and used fluorescent protein reporters fused to a weak ssrA degradation tag (AANDENYAASV). Fresh cultures were grown from single colonies streaked on LB agar the previous day from a glycerol stock frozen at −80°C. These inoculum cultures were grown out for 6 h in 3 ml of LB media at 37°C in 14-ml culture tubes and then diluted into 3 ml of MM containing 0.8% glycerol in 14-ml culture tubes for 18 h of overnight growth. The initial dilution of these overnight cultures was carefully set to a sufficiently low $OD_{600}$ so that the cell density after 18 h of growth would be below an $OD_{600}$ of 0.5 so as not to induce the oxygen sensor. The overnight culture was diluted to an $OD_{600}$ of 0.01 in pre-warmed 30 ml of MM containing carbon sources as indicated. Cultures were grown at 37°C at 250 RPM in an Innova 44 incubator (New Brunswick) with a circular throw diameter of 1 inch. Samples of 1 ml were removed at different time points and their $OD_{600}$ was immediately measured using a Cary 50 UV-vis spectrophotometer (Varian) before freezing at −20°C in a 96-well deep-well plate for later analysis.

### Cytometry

Cells were analyzed by flow cytometry on a LSR Fortessa analyzer (BD) with a 488-nm laser and 510/20-nm band pass filter to collect GFP fluorescence and a 561-nm laser and 610/20-nm band pass filter to collect RFP fluorescence. Cell samples were diluted below $OD_{600}$ of 0.01 in PBS to ensure separation of cell events. Cell samples were analyzed by a High-Throughput Sampler at a flow rate of 0.5 µl/s until $10^4$–$10^5$ gated counts were collected. FSC-H and SSC-H thresholds were set to exclude background events. Data were analyzed using FlowJo software (Treestar). The median of the fluorescence histogram of each gated population was calculated and is reported here as the fluorescence value of a sample in arbitrary units (a.u.).

### Dissolved oxygen measurements

To measure dissolved oxygen (DO) during shake flask growth, we used a FireStingO$_2$ oxygen sensor (PyroScience) with an OXF1100 needle. The sensor was calibrated to 100% DO (210 µmol/l) with a single point calibration in MM heated to 37°C and vigorously shaken for 5 min prior to calibration. DO measurements were taken by submerging the sensor needle in the culture immediately after removing the culture flask from the incubator. Sensor readings were recorded in real time in the FireStingO$_2$ software and were analyzed for equilibrium DO concentrations. Readings below 20 µmol/l could not be accurately attained; therefore, lower readings are reported at this lower limit. Following DO measurement, 1 ml of each culture was sampled to measure cell density ($OD_{600}$) on a Cary 50 UV-vis spectrophotometer (Varian).

### Liquid chromatography

Culture samples frozen at −20°C in 96-well plates were thawed in a 42°C water bath for 30 min. Sample plates were then centrifuged at 4,255 *g* for 10 min three times, each time pipetting half the supernatant into a clean plate. After the third centrifugation,

100 µl of supernatant was pipetted into a U-bottom 96-well plate (Corning #3797). This plate was covered with an AluminaSeal (Diversified Biotech #ALUM-1000) and kept at 4°C until analysis. Supernatant analysis was performed using an Agilent 1260 Infinity Liquid Chromatography system with an inline Aminex HPX-87H column (#125-0140) and Micro-Guard Cation column (Bio-Rad #125-0129) running a 5 mM sulfuric acid mobile phase at 0.6 ml/min. Purified supernatant samples in 96-well plates were placed in an autosampler cooled to 4°C. Following sampling of 10 µl, the autosampler needle was cleaned with a 3-s rinse of 10% isopropanol. The peaks for glucose (9.2 min) and acetate (15.5 min) were detected with a Refractive Index detector (RID; Agilent #G1362A). Both the columns and the RID were heated to 35°C. Standard curves of glucose (Sigma #049K6201) and acetate (Fluka #57191) were run to enable quantification. Integration of the RID peaks for glucose and acetate was done automatically in Chemstation software (Agilent).

### Modeling

To generate circuit models, the sensors' promoter activities were first converted to relative promoter units (RPU) by multiplying the background-subtracted fluorescence levels by a conversion factor ($10^{-3}$) estimated such that the units are comparable to a previously published standard (Nielsen *et al*, 2016). The minimum and maximum values for each sensor's output promoter are (in RPU) as follows: glucose, 0.006–0.237; low oxygen, 0.020–1.346; and acetate 0.002 and 0.700. The output of each sensor was measured at discrete 1-h time points over 27-h growth experiments. For the purposes of the simulation, sensor outputs in between measurements are determined using a linear interpolation. All possible 3-input, 1-output truth tables were designed by Cello (version 1.0 with Eco1C1G1T1 UCF) using the minimum and maximum RPU values for the sensors (Nielsen *et al*, 2016). The output of Cello includes a wiring diagram of NOR/NOT gates that produces the desired truth table as well as the specific repressors assigned to each gate. The circuits based on AND and ANDN gates were designed by hand. Cello only predicts the steady-state behavior of the circuit. Simple dynamic simulations were run to evaluate how the circuits respond to changes in the sensors over time. The steady-state response function for each NOR/NOT gate is captured by

$$y = y_{\min} + (y_{\max} - y_{\min})\frac{K^n}{K^n + x^n}, \tag{1}$$

where $x$ is the activity of the input promoter (for a NOR gate, $x$ is the sum of the input promoters $x = x_1 + x_2$), $y$ is the output of the gate, and the parameters are dependent on the assigned repressor and have been published previously (Nielsen *et al*, 2016). For the AND gate, the response function is

$$y = y_{\min} + (y_{\max} - y_{\min})\frac{x_1 x_2{}^2}{K + x_1 x_2{}^2}, \tag{2}$$

where $x_1$ and $x_2$ are the outputs from glucose and acetate sensors, respectively, and the parameters are estimated to be $y_{\min} = 0.001$ RPU, $y_{\max} = 0.3$ RPU, and $K = 10^{-5}$ RPU. For the ANDN gate,

$$y = y_{\min} + (x_1 - y_{\min}) \frac{K}{K + x_2}, \tag{3}$$

where $x_1$ and $x_2$ are the outputs from glucose and oxygen sensors, respectively, $y_{\min} = 0.001$ RPU, and $K = 0.0025$ RPU. To simulate the dynamic response of each circuit, a set of ordinary differential equations (ODEs) is solved, where each ODE represents the change in the output activity of a gate in the circuit according to

$$\frac{\mathrm{d}y}{\mathrm{d}t} = \alpha (y_{\max} - y_{\min}) \frac{K^n}{K^n + x(t)^n} - \gamma (y(t) - y_{\min}), \tag{4}$$

where $\alpha$ and $\gamma$ are the rate constants for turning a gate ON and OFF, respectively. Parameters $y_{\min}$, $y_{\max}$, $K$, and $n$ are the same as equation (1), and $\alpha$ and $\gamma$ are estimated to be 1 per hour (Tabor *et al*, 2009; Moon *et al*, 2012). Equations (5–7) show the complete set of equations for an example 3-input ($x_1$, $x_2$, $x_3$) and 1-output ($y_3$) circuit (Circuit B in the top panel of Appendix Fig S8):

$$\frac{\mathrm{d}y_1}{\mathrm{d}t} = \alpha (y_{\max,1} - y_{\min,1}) \frac{K_1^{n_1}}{K_1^{n_1} + [x_1(t) + x_2(t)]^{n_1}} - \gamma (y_1(t) - y_{\min,1}), \tag{5}$$

$$\frac{\mathrm{d}y_2}{\mathrm{d}t} = \alpha (y_{\max,2} - y_{\min,2}) \frac{K_2^{n_2}}{K_2^{n_2} + x_3(t)^{n_2}} - \gamma (y_2(t) - y_{\min,2}), \tag{6}$$

$$\frac{\mathrm{d}y_3}{\mathrm{d}t} = \alpha (y_{\max,3} - y_{\min,3}) \frac{K_3^{n_3}}{K_3^{n_3} + [y_1(t) + y_2(t)]^{n_3}} - \gamma (y_3(t) - y_{\min,3}) \tag{7}$$

This system of ODEs is solved discretely in Python for an interval of 27 h, using a time step size of 0.025 h. In each time step, the corresponding empirical values for the output activity of glucose, oxygen, and acetate sensors are assigned to the inputs $x_1$, $x_2$, and $x_3$, respectively. The initial conditions for $y_1$, $y_2$, and $y_3$ are also determined by solving the above system of ODEs at steady state using the following sensor output activities: $x_1 = 1.294$ RPU, $x_2 = 0.006$ RPU, and $x_3 = 0.028$ RPU.

### RNA-seq library preparation

RNA-seq library preparation and sequencing were performed following the methods described in Gorochowski *et al* (2017). Briefly, total RNA was harvested from *E. coli* MG1655 *ΔglnL* cells at different time points specified above. Cells were grown in minimal media containing 1.6% glucose. At least 2 million cells were collected at each time point, as assessed by the culture's $OD_{600}$. This was done by spinning down sufficient volume of each culture at 4°C and 15,000 × *g* for 3 min, discarding the supernatant, and flash freezing the cell pellets in liquid nitrogen for storage at −80°C. After lysing the cells with 1 mg of lysozyme (Sigma-Aldrich, MO, L6871) in 10 mM Tris–HCl (pH 8.0) with 0.1 mM EDTA (USB 75825 and 15694, respectively), RNA was extracted with PureLink RNA Mini Kit (Life Technologies, CA, 12183020). RNA Clean & Concentrator-5 (Zymo Research, R1015) was used to further purify and concentrate the RNA samples, verified by

Bioanalyzer (Agilent, CA). Next, ribosomal RNAs were depleted from the samples using Ribo-Zero rRNA Removal Kit for bacteria (Illumina, CA, MRZMB126). Only samples with RNA integrity number (RIN) > 8.5 were considered for the subsequent library preparation steps. Strand-specific RNAtag-seq libraries were created by the Broad Technology Labs specialized service facility (SSF) (Gorochowski *et al*, 2017). Each sample was tagged with a unique barcode, and all samples were pooled together to run on two separate lanes of an Illumina HiSeq 2500. Finally, sequencing reads were generated by re-pooling the reads from the two lanes, de-multiplexing them into the original samples, and trimming the barcode tag from each read.

### Processing of sequencing data

Alignment of raw reads and transcription profile generation were performed following a previously developed in-house Python script (Gorochowski *et al*, 2017). Briefly, raw reads were mapped to the genome of *E. coli* MG1655 (NCBI RefSeq: NC_000913.3), with one modification in which the relevant region around *glnL* gene was deleted to yield *E. coli* MG1655*ΔglnL*. The alignment of raw reads was done using BWA version 0.7.4 with default settings (Li & Durbin, 2009), followed by generating the corresponding SAM files and BAM files. Next, the sense and anti-sense transcription profiles were generated by identifying the position of mapped reads in the forward and reverse directions, respectively. To do that, the BAM files were filtered using the "view" command of SAMtools and the sense reads were selected using the filter codes 83 and 163, and anti-sense reads were selected using filter codes 99 and 147. The normalized FPKM values were calculated by averaging the height of transcription profile along the length of each gene, normalized by the total mapped nucleotides across the genome, and multiplied by $10^9$. To account for potential variations introduced during library preparation, between-sample normalization factors were calculated using the trimmed mean of M-values approach (TMM; Robinson & Oshlack, 2010), and were applied to the FPKM values in each sample.

## Data availability

Full annotated plasmid sequences, code for generating promoter libraries, and detailed descriptions of promoter sequences are provided in the authors' Github repository (https://github.com/VoigtLab/promoter-library-design-tool). Additional data is available upon request.

**Expanded View** for this article is available online.

### Acknowledgements

This work was supported by US National Science Foundation Synthetic Biology Engineering Research Center (SynBERC EEC0540879) and the Office of Naval Research Multidisciplinary University Research Initiative (N00014-13-1-0074) and the Department of Energy (DE-SC0018368, DE-SC0018368).

### Author contributions

FM and CAV conceived of the study. FM designed and performed the experiments and analyzed the data. AEB did the modeling and analyzed RNA seq

data. ANG designed the sRNA system. EC designed the self-targeting *mf*-LON system. YP handled samples for the RNA seq experiment. FM, AEB, and CAV wrote the manuscript.

## Conflict of interest

The author declares that they have no conflict of interest.

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
