## [Review Process File · Molecular Systems Biology]

Dynamic control of endogenous metabolism with combinatorial logic circuits

Moser, F, Espah Borujeni, A, Ghodasara, A, Cameron, E, Park, Y-J, and C.A. Voigt.

Review timeline:

Submission date:	10 th September 2018
Editorial Decision:	1 st October 2018
Revision received:	25 th October 2018
Accepted:	30 th October 2018

Editor: Maria Polychronidou

Transaction Report:

1st Editorial Decision

1st October 2018

Thank you again for submitting your work to Molecular Systems Biology. We have now heard back from the three referees who agreed to evaluate your study. As you will see below, the reviewers think that the study is interesting and novel and they acknowledge the quality of the presented data and analyses. They raise however a series of relatively minor issues, which we would ask you to address in a minor revision.

Overall, I think that the reviewers' recommendations are clear and there is therefore no need to repeat the points listed below. Please let me know in case you would like to discuss in further detail any of the issues raised by the reviewers.

REFeree REPORTS.

Reviewer #1:

The manuscript by Moser et al. describes an approach where the sensing of internal cues within a cell is interfaced with E.coli modular logic gate systems previously developed by the Voigt lab. The fact that the cues do not arrive at the same time in an E.coli growth cycle means that the logic systems behave with sequential behaviour. This provides a route to programming automatic differential regulation during an E.coli growth cycle without needing to add any external inducers. The authors use this approach to control native E.coli metabolism to prevent acetate accumulation by the end of a growth cycle. The work is ambitious and novel, and likely to be of great interest to your journal as it nicely combines mathematical biology, synthetic biology and metabolic engineering.

Furthermore, it is very well written and well presented and amongst the main story it also nicely describes some use of useful new methods - promoters are designed and screened with a microarray-based synthesis approach, and promoters are later characterised during a growth cycle by RNAseq.

It also has a very useful and substantial introduction, with a large number of citations too.

I recommend this work for publication, with only a few minor points which I think the authors should consider revising or addressing before public release.

Minor points.

1. In previous work the authors mention NAND gates but here they are termed ANDN gates. I assume (maybe incorrectly) that they are the same thing, and if so I think it would be best to keep to NAND.

2. The final line of the introduction says that the 'sensors and gates can be reconfigured to respond at different times' for the pathway example. I don't think that this was actually shown in the manuscript. My interpretation of this sentence is that with the same sensors and the same output genes, using different logic in the middle part of the network can change the dynamical response. This is true for the Fig 2 work but not the Fig 4 work, where the outputs of the circuitry are completely different for the two designs shown. I would consider changing this sentence so that the claims and results match better.

3. Figure 3C - At first glance there doesn't appear to be any benefit for adding CRISPRi as well as mF-Lon but then I noticed that the y-axis scales were different. I think it would be best to keep all the y-axis shown to the same scale.

4. Figure 4B & 4E - it would be good to also see growth curves in the Supporting info so we can see how the system dynamics tally with the growth phases and any negative effect on growth that is imparted by the system

5. Figure S8 - this is pretty hard to deconvolute. It's certainly impressive but not very reader friendly. I would suggest spreading this data over several figures rather than all-in-one.

Reviewer #2:

In this manuscript, Moser, et al. present results on the construction and design of logic gates to implement temporal control of endogenous genes based on sensed indicators of growth phase-dependent properties (glucose, dissolved oxygen, and acetate). My impression is that a heroic amount of work went into the data in this manuscript. I am enthusiastic about the overall goal of developing accurate sensors that can be used to build logic gates for temporal control of metabolic flux. This is an exciting area where there have been a number of recent advances and this contribution complements them nicely. In the end, it was not entirely clear to me that the full suite of tools deployed would be easy to implement in other settings (e.g. limits on toxicity and stability) or would be strictly necessary (e.g. sgRNA designs are often comparable to more complex systems with the protease). However, overall I found the work to be of high quality and it certainly offers an additional angle on dynamic control strategies for controlling metabolic processes. I have a number of small corrections, requests for controls, and comments about ways to improve clarity.

Minor comments

1. In the text discussion Figs. 3b, c the authors claim that "dual control outperformed on both the speed of the response and the potency of the knockdown." Please perform statistical tests to show that there is indeed a difference in each of these cases. This comment extends to Fig. 3d, where it is unclear visually that the combination of sgRNA and protease is any different than sgRNA alone.

2. The P_{glnA}P_{2s} acetate sensor is only functional in a strain with *glnL* deleted, and all future experiments are done using this deletion. Please include discussion on whether there are drawbacks to having this deletion required.

3. The caption of Fig. S5 mentions Fig. 1J, which doesn't exist.

4. Fig. S5 mentions P_{dexA7}, which does not appear elsewhere in the paper and is perhaps a historical name.

5. In Fig. S9b it appears that the AND gate function is transient. What are the implications for the circuit function at longer time scales?

6. Fig. S9 please indicate what time scale was used.

7. In Fig. 2f, I was confused about the promoter regulating rfp. It is a copy of the PgluA7 promoter, but there is also regulation by PhIF, if I understand correctly. It would be helpful to include additional information to clarify what this promoter architecture looks like.
8. In Table S1 it would be helpful to have the final OD or some other measure of total colony counts in addition to the growth rate data.
9. For the data in Fig. 4 the authors state that "These data confirmed that poxB expression peaks during the transition from exponential to stationary phase." It would be useful to include the growth curve data somewhere.
10. In Fig. 4c and its caption, WT corresponds to Δ glnL, Circuit (-tag/sgRNA) corresponds to Δ glnL Δ pta poxB::E170. Why is the acetate production of these two the same, while in Fig. S14, the latter one should only be 50% of the WT? In the text that describes this figure, parent cells are Δ glnL Δ pta poxB::E170. Does this mean that this is actually the WT and the figure caption is incorrect?
11. The colored bars showing glucose, etc. levels are helpful, but I was unable to find a description for how they were determined. In particular, the bar positions vary between Fig. 4b and e and those in Fig. 2c so it would be useful to know how they were obtained.
12. In Fig. S4, please add the result for Δ glnL Δ pta Δ poxB since this strain is important for the applications later in the manuscript.
13. In addition to Fig. 1b, it would be helpful to include the base DNA sequence information used somewhere in supplementary with additional details on what the random spacer sequences are, the locations of the operators, etc.
14. Should the error bars in Fig. 1g be horizontal? For the Fig. 1f and h the error bars correspond to GFP measurements.
15. The binding site location is listed in the figure for Fig. 1f and g, but not for NRI in Fig. 1h.
16. pg. 9 ".....can be detrimental under these conditions. 115-12" looks like it might be an error in reference formatting.
17. The reference to Figure S14 at the top of page 10 should be to Figure S15 instead.

Reviewer #3:

Comments for Author:

Review for "Dynamic control of endogenous metabolism with combinatorial logic circuits"

In this manuscript, the authors explore the use of a combination of de novo constructed glucose and oxygen sensors and an optimized acetate sensor in a logic circuit to provide temporal responses in dynamic (batch) culture conditions. These sensors were used in various logic circuits, using GFP and RFP readouts to demonstrate the ability for temporal responses of each circuit. Finally, the authors combined targeted proteolysis with CRISPRi to enable rapid modulation of pta and poxB protein levels. This strategy was subsequently integrated into two genetic sensor circuits to demonstrate acetate formation can be dynamically regulated based on changing external factors (glucose, oxygen and acetate levels) experienced by cell factories in a batch cultivation. After reading the manuscript, my opinion is that this impressive work is suitable for publication in Molecular Systems Biology. I have no major comments and summarized my minor comments and suggestions point-by-point below.

Minor comments:

- Page 10: "This is predicted to be on during stationary phase and turn off as cells transition to stationary phase, which mimics when pta is transcribed (Figure 4e)." The first mention of stationary phase should be changed to exponential phase.
- Page 11: "flocculation for sedimentation for biomass removal inhibition of cell growth" should be changed to "removal and inhibition"
- When describing medium with glycerol as the carbon source, the authors should indicate if the "%" describes weight % (g/g), volume % (ml/ml) or weight/volume (g/ml).

- Page 15: As excluding trace amounts of oxygen of in anaerobic chambers is challenging and could impact the measured off-state of the oxygen sensor, the authors should include additional details regarding their anaerobic setup (e.g. brand of the chamber, catalyst, regeneration of the catalyst).

- Page 15: The authors mention that the anaerobic cultures were grown for an additional 6 hours prior to sampling, the authors should motivate why these extra hours of incubation were performed/necessary.

- Page 15: "Oxygen sensors and oxygen modulated circuits were tested in 14 ml culture tubes, unless otherwise noted, in order to achieve anaerobic conditions." This sentence is not clear, why does testing in 14 ml culture tubes result in anaerobic conditions? Anaerobic conditions were achieved due to the use of the rubber stopper and flushing with nitrogen, mentioned later in the text.

- Page 16: "Fresh cultures were inoculated from fresh single colonies streaked on LB agar from a glycerol stock frozen at -80 {degree sign}C." What is defined as a "fresh single colony"? Also the first "fresh" can be removed from this sentence. Finally, are the cultures mentioned by the authors here the inoculum cultures? If so, this should be clarified.

Suggestions:

- Page 3: "However, an individual sensor can only implement a switch at a one defined time and cannot be used to drive a series of events." Changing "defined time" by "defined metabolic state" would be a more accurate description.

- Page 14: "For oxygen sensors, cells were grown in 0.4% glycerol in either aerated tubes or tubes from which oxygen had been removed (see below)" Changing to "MM containing 0.4% glycerol" would make the interpretation less ambiguous.

- Lay-out of the references could be improved (e.g. removal of excess capitals and using italics for the appropriate words)

1st Revision - authors' response

25th October 2018

Reviewer #1:

1. In previous work the authors mention NAND gates but here they are termed ANDN gates. I assume (maybe incorrectly) that they are the same thing, and if so I think it would be best to keep to NAND.

NAND and ANDN are different gate types. We have now clarified this in the text.

2. The final line of the introduction says that the 'sensors and gates can be reconfigured to respond at different times' for the pathway example. I don't think that this was actually shown in the manuscript. My interpretation of this sentence is that with the same sensors and the same output genes, using different logic in the middle part of the network can change the dynamical response. This is true for the Fig 2 work but not the Fig 4 work, where the outputs of the circuitry are completely different for the two designs shown. I would consider changing this sentence so that the claims and results match better.

The sentence has been edited for clarity.

3. Figure 3C - At first glance there doesn't appear to be any benefit for adding CRISPRi as well as mF-Lon but then I noticed that the y-axis scales were different. I think it would be best to keep all the y-axis shown to the same scale.

We have changed all the Y-axes in Figure 3C to the same scale.

4. Figure 4B & 4E - it would be good to also see growth curves in the Supporting info so we can see how the system dynamics tally with the growth phases and any negative effect on growth that is imparted by the system

We have added full growth curves of the strains in this figure as Appendix Figure S16.

5. Figure S8 - this is pretty hard to deconvolute. It's certainly impressive but not very reader friendly. I would suggest spreading this data over several figures rather than all-in-one.

The intention of the figure is to display the diversity of responses that can be obtained by using different circuit architectures, which easiest to see when presented together. We have modified the figure to help the reader visually separate the component figures.

Reviewer #2:

1. In the text discussion Figs. 3b, c the authors claim that "dual control outperformed on both the speed of the response and the potency of the knockdown." Please perform statistical tests to show that there is indeed a difference in each of these cases. This comment extends to Fig. 3d, where it is unclear visually that the combination of sgRNA and protease is any different than sgRNA alone.

We have performed and included the requested statistical tests for these Figures. It is true, however, that we could not detect a statistically significant ($P < 0.01$) difference between the single knockdown and the combination knockdown system in terms of the fold-knockdown. Rather, this configuration combines the speed of the protease with the fold-change of CRISPRi. The claims have been edited to reflect this.

2. The *PglnAP2s* acetate sensor is only functional in a strain with *glnL* deleted, and all future experiments are done using this deletion. Please include discussion on whether there are drawbacks to having this deletion required.

We have addressed the requirement of this mutation in the text.

3. The caption of Fig. S5 mentions Fig. 1J, which doesn't exist.

This has been corrected to say Figure 1e.

4. Fig. S5 mentions *PdexA7*, which is does not appear elsewhere in the paper and is perhaps a historical name.

That was indeed a historical name and have corrected it in the figure.

5. In Fig. S9b it appears that the AND gate function is transient. What are the implications for the circuit function at longer time scales?

We do observe a decrease in the output of this AND gate at later times and attribute this to the consumption of glucose late in growth. While the signal is attenuated, it never reverts to its baseline state at the measured time scales and therefore can still complete its AND function.

6. Fig. S9 please indicate what time scale was used.

We have indicated that time intervals are one hour between cytometry histograms. Circuit performance was tested as in Figure 2g,j and described in the *Fluorescence Assays* section of the Methods.

7. In Fig. 2f, I was confused about the promoter regulating *rfp*. It is a copy of the *PgluA7* promoter, but there is also regulation by *PhlF*, if I understand correctly. It would be helpful to include addition information to clarify what this promoter architecture looks like.

We have clarified the description of this promoter in the figure legend and added the complete sequence of the PhIF-repressed promoter to Appendix Table S2.

8. *In Table S1 it would be helpful to have the final OD or some other measure of total colony counts in addition to the growth rate data.*

We have added the final density (OD600) of these cultures to Table S1.

9. *For the data in Fig. 4 the authors state that "These data confirmed that *poxB* expression peaks during the transition from exponential to stationary phase." It would be useful to include the growth curve data somewhere.*

We have added the complete growth curves from Figure 4 as Appendix Figure S16.

10. *In Fig. 4c and its caption, WT corresponds to Δ glnL, Circuit (-tag/sgRNA) corresponds to Δ glnL Δ pta *poxB*::E170. Why is the acetate production of these two are the same, while in Fig. S14, the latter one should only be 50% of the WT? In the text that describes this figure, parent cells are Δ glnL Δ pta *poxB*::E170. Does this mean that this is actually the WT and the figure caption is incorrect?*

"WT" in Fig 4c supposed to be MG1655 Δ glnL Δ pta *poxB*::E170 (no plasmids). "WT" in Figure 4f is supposed to be Δ glnL Δ poxB pta::pdt3. We have corrected the figure labels and legend accordingly.

11. *The colored bars showing glucose, etc. levels are helpful, but I was unable to find a description for how they were determined. In particular, the bar positions vary between Fig. 4b and e and those in Fig. 2c so it would be useful to know how they were obtained.*

The colored bars represent when the sensors are active, measured empirically as the activity of their output promoters under the conditions of the growth experiments. The reason these shift between Figure 2c and Figure 4b is because the growth rate of the MG1655 Δ glnL Δ pta *poxB*::E170 strain in Figure 4b is slower than that of the MG1655 Δ glnL strain we used for Figure 2c. In Figure 4b, the glucose and acetate curves are therefore shifted to the right and overlap for longer. We have clarified in the text when the sensors are considered active and note the difference between the figures.

12. *In Fig. S4, please add the result for Δ gln Δ pta Δ poxB since this strain is important for the applications later in the manuscript.*

We have added additional results to Figure S4 for the behavior of PglNAP2 in both the strain MG1655 Δ gln Δ pta Δ poxB and MG1655 Δ gln Δ poxB.

13. *In addition to Fig. 1b, it would be helpful to include the base DNA sequence information used somewhere in supplementary with additional details on what the random spacer sequences are, the locations of the operators, etc.*

We have submitted the Matlab code we used and a document that details the exact sequences we used and where the operators were placed Github (Supplemental Material). In addition, we have added the random spacer sequences to Appendix Table S2.

14. *Should the error bars in Fig. 1g be horizontal? For the Fig. 1f and h the error bars correspond to GFP measurements.*

Yes, the horizontal error bars reflect measurement errors during dissolved oxygen measurements. We have clarified this in the figure legend.

15. *The binding site location is listed in the figure for Fig. 1f and g, but not for NRI in Fig. 1h.*

We have added the locations of each NRI binding sites in the PglNAP2 promoter in Fig 1.

16. pg. 9 ".....can be detrimental under these conditions. 115-12" looks like it might be an error in reference formatting.

We have corrected the citation.

17. The reference to Figure S14 at the top of page 10 should be to Figure S15 instead.

We have corrected this typo.

Reviewer #3:

1. Page 10: "This is predicted to be on during stationary phase and turn off as cells transition to stationary phase, which mimics when *pta* is transcribed (Figure 4e)." The first mention of stationary phase should be changed to exponential phase.

We have made the correction.

2. Page 11: "flocculation for sedimentation for biomass removal inhibition of cell growth" should be changed to "removal and inhibition"

We have made the correction.

3. When describing medium with glycerol as the carbon source, the authors should indicate if the "%" describes weight % (g/g), volume % (ml/ml) or weight/volume (g/ml).

We have added a line on Page 13 under Media to clarify that all % is in terms of % mass (g/g).

4. Page 15: As excluding trace amounts of oxygen of in anaerobic chambers is challenging and could impact the measured off-state of the oxygen sensor, the authors should include additional details regarding their anaerobic setup (e.g. brand of the chamber, catalyst, regeneration of the catalyst).

We have included additional details as requested in the Methods.

5. Page 15: The authors mention that the anaerobic cultures were grown for an additional 6 hours prior to sampling, the authors should motivate why these extra hours of incubation were performed/necessary.

We have added further clarification on page 15 in the Methods. Briefly, the cells were grown for an additional 6 hours to enable the output GFP to be produced.

6. Page 15: "Oxygen sensors and oxygen modulated circuits were tested in 14 ml culture tubes, unless otherwise noted, in order to achieve anaerobic conditions." This sentence is not clear, why does testing in 14 ml culture tubes result in anaerobic conditions? Anaerobic conditions were achieved due to the use of the rubber stopper and flushing with nitrogen, mentioned later in the text.

We have added further clarification in the Methods. The 14 ml tubes were necessary because we used rubber stoppers to seal the tubes, which prevented further oxygen from entering the tube following the vacuum/nitrogen flushing.

7. Page 16: "Fresh cultures were inoculated from fresh single colonies streaked on LB agar from a glycerol stock frozen at -80{degree sign}C." What is defined as a "fresh single colony"? Also the first "fresh" can be removed from this sentence. Finally, are the cultures mentioned by the authors here the inoculum cultures? If so, this should be clarified.

We have added further clarification to address these questions. With "fresh" colony, we meant one that was streaked the previous day. We have clarified that these cultures are inoculum.

Suggestions:

- Page 3: *"However, an individual sensor can only implement a switch at a one defined time and cannot be used to drive a series of events." Changing "defined time" by "defined metabolic state" would be a more accurate description.*

We have incorporated the suggestion.

- Page 14: *"For oxygen sensors, cells were grown in 0.4% glycerol in either aerated tubes or tubes from which oxygen had been removed (see below)" Changing to "MM containing 0.4% glycerol" would make the interpretation less ambiguous.*

We have incorporated the suggestion.

- *Lay-out of the references could be improved (e.g. removal of excess capitals and using italics for the appropriate words)*

We have incorporated the suggestion.

Accepted

30th October 2018

Thank you again for sending us your revised manuscript. We are now satisfied with the modifications made and I am pleased to inform you that your paper has been accepted for publication.

MOLECULAR SYSTEMS BIOLOGY

Corresponding Author Name: Christopher Voigt
Manuscript Number: MSB-18-8605